Manuscript prepared for Atmos. Meas. Tech.
with version 2014/05/30 6.91 Copernicus papers of the LaTeX class copernicus.cls.
Date: 25 May 2017

# MIPAS IMK/IAA Carbon Tetrachloride (CCl$_4$) Retrieval and first Comparison with other Instruments

**E. Eckert**[1], **T. von Clarmann**[1], **A. Laeng**[1], **G. P. Stiller**[1], **B. Funke**[1], **N. Glatthor**[1], **U. Grabowski**[1], **S. Kellmann**[1], **M. Kiefer**[1], **A. Linden**[1], **A. Babenhauserheide**[1], **G. Wetzel**[1], **C. Boone**[2], **A. Engel**[3], **J. J. Harrison**[4,5,6], **P. E. Sheese**[7], **K. A. Walker**[2,7], **and P. F. Bernath**[2,8]

[1]Karlsruhe Institute of Technology, Institute of Meteorology and Climate Research, Karlsruhe, Germany

[2]Department of Chemistry, University of Waterloo, Waterloo, Ontario, Canada

[3]Institut für Atmosphäre und Umwelt, J. W. Goethe Universität, Frankfurt, Germany

[4]Department of Physics, University of Leicester, University Road, Leicester LE1 7RH, United Kingdom

[5]National Centre for Earth Observation, University of Leicester, University Road, Leicester LE1 7RH, United Kingdom

[6]Leicester Institute for Space and Earth Observation, University of Leicester, University Road, Leicester LE1 7RH, United Kingdom

[7]Department of Physics, University of Toronto, Toronto, Ontario, Canada

[8]Department of Chemistry and Biochemistry, Old Dominion University, Norfolk, VA 23529-0126, USA

*Correspondence to:* E. Eckert (ellen.eckert@kit.edu)

**Abstract.** MIPAS thermal limb emission measurements were used to derive vertically resolved profiles of carbon tetrachloride (CCl$_4$). Level-1b data versions MIPAS/5.02 to MIPAS/5.06 were converted into volume mixing ratio profiles using the level-2 processor developed at Karlsruhe Institute of Technology (KIT) Institute of Meteorology and Climate Research (IMK) and Consejo Superior de Investigaciones Científicas (CSIC), Instituto de Astrofísica de Andalucía (IAA). Consideration of peroxyacetyl nitrate (PAN) as interfering species, which is jointly retrieved, and CO$_2$ line mixing is crucial for reliable retrievals. Parts of the CO$_2$ Q-branch region that overlap with the CCl$_4$ signature were omitted, since large residuals were still found even though line mixing was considered in the forward model. However, the omitted spectral region could be narrowed noticeably when line mixing was accounted for. A new CCl$_4$ spectroscopic dataset leads to slightly smaller CCl$_4$ volume mixing ratios. In general, latitude-altitude cross-section show the expected CCl$_4$ features with highest values of around 90 pptv at altitudes at and below the tropical tropopause and values decreasing with altitude and latitude due to stratospheric decomposition. Other patterns, such as subsidence in the polar vortex during winter and early spring, are also visible in the distributions. The decline in CCl$_4$ abundance during the MIPAS Envisat measurement period (July 2002 to April

2012) is clearly reflected in the altitude-latitude cross-section of trends estimated from the entire retrieved data set.

## 1 Introduction

Carbon tetrachloride (CCl$_4$) is an anthropogenically produced halogen yielding trace gas and partly responsible for stratospheric ozone depletion. It is also a potent greenhouse gas with a 100-year global warming potential of 1730 (IPCC, 2013; World Meteorological Organization (WMO), 2014). CCl$_4$ was commonly used in fire extinguishers, as a precursor to refrigerants and in dry cleaning prior to 1990, when it was restricted within the framework of the London Amendment to the Montreal Protocol. Its abundances in the atmosphere increased steadily from the first part of the 20th century. Emissions declined significantly after 1990, as well as the amount of CCl$_4$ in the atmosphere with a few years delay. 2007-2012 bottom-up emssions of 1 to 4 kilotonnes/year were assessed by combining country-by-country reports to the United Nations Environmental Programme (UNEP) (Liang et al., 2016). This bottom-up estimate differs considerably from the 57($\pm$17) kilotonnes/year top-down emissions which were evaluated in 2014 (World Meteorological Organization (WMO), 2014) using atmospheric measurements and lifetime estimates. Even

when possible $CCl_4$ precursors and unreported, inadvertent emissions are accounted for, the gap between reported bottom-up and estimated top-down $CCl_4$ emissions cannot be closed, as bottom-up emissions still only add up to 25 kilotonnes/year (SPARC, 2016). Besides a sink in the atmosphere, $CCl_4$ is decomposed in the ocean and the soil with different lifetimes for each sink. Reassessment of the different lifetime estimates, which are essential for an adequate top-down assessment of emissions, leads to lower emissions of $\sim$40($\pm$15) kilotonnes/year. While the gap between bottom-up and top-down emissions is now smaller after reassessments, the discrepancy is still not solved entirely. Previous measurements of stratospheric $CCl_4$ have also been performed by the Atmospheric Chemistry Experiment Fourier Transform Spectrometer (ACE-FTS), a Cryosampler instrument employed at Frankfurt University and the balloon borne version of MIPAS (MIPAS-B2). The first version of the balloon borne MIPAS instrument (MIPAS-B) and ATMOS (Atmospheric Trace Molecule Spectroscopy) also measured $CCl_4$, but not during the MIPAS Envisat measurement period (Zander et al., 1996; von Clarmann et al., 1995).

Additional measurements, especially vertically well resolved ones with global coverage such as satellite measurements from MIPAS, can help to improve the understanding of the atmospheric $CCl_4$ budget and stratospheric lifetime estimate. Furthermore, as a tracer with relatively short stratospheric lifetimes, $CCl_4$ measurements can improve the understanding of changes in Brewer-Dobson circulation further constraining the lower boundary, e.g. processes around the tropopause.

In this study, we present the retrieval of $CCl_4$ distributions from MIPAS limb emission spectra. First, we characterize the MIPAS instrument (Sec. 2), followed by a detailed description of the retrieval and the specific issues that had to be dealt with to derive $CCl_4$ concentration (Sec. 3). We then compare the results of the MIPAS Envisat $CCl_4$ retrieval with those of ACE-FTS, those of the second balloon-borne MIPAS instrument (MIPAS-B2) and those of Cryosampler measurements (Sec. 5) and summarize the results in the conclusions (Sec. 6).

## 2   MIPAS

The Michelson Interferometer for Passive Atmospheric Sounding (MIPAS) was one of the instruments aboard the European Environmental Satellite (Envisat). It was launched into a sun-synchronous orbit at an altitude of approximately 800 km on 1 March 2002. On 8 April 2012, all communication with the satellite was lost ending an observation period of more than 10 years. Envisat orbited the earth 14.4 times a day crossing the equator at 10:00 and 22:00 local time. MIPAS measured infrared emissions between 685 $cm^{-1}$ and 2410 $cm^{-1}$ (14.6 and 4.15µm) (Fischer et al., 2008), which

allows for day and night time measurements with global coverage. The initial spectral resolution of the instrument was 0.025 $cm^{-1}$ (0.0483 $cm^{-1}$ after a "Norton-Beer strong" apodization (Norton and Beer, 1976)). An instrument failure in March 2004 led to an observation gap until January 2005 when the instrument was successfully restarted. The first period (June 2002 to March 2004) is referred to as full spectral resolution (FR) period, while the period from January 2005 to April 2012 is referred to as reduced spectral resolution (RR) period. Due to a problem with one of the interferometer slides, MIPAS could only be operated with a spectral resolution of 0.0625 $cm^{-1}$ (0.121 $cm^{-1}$ apodized) from January 2005 on. In this study, only measurements from the instrument's "nominal operation mode" are used. In this mode, the number of tangent altitudes increased from 17 during the FR period to 27 during the RR period. The vertical coverage ranges from 6 km to around 68 km during the FR period and up to around 70 km during the RR period, respectively. MIPAS initially took around 1000 measurements per day. In 2005, operation was resumed at reduced duty cycle. By the end of 2007, MIPAS was back at full duty cycle which amounts to approximately 1300 RR measurements per day. The horizontal sampling changed from 510 km during the FR period to 410 km during the RR period.

The temperature and various atmospheric trace gases are retrieved from level-1b data using a retrieval processor developed at the Institute of Meteorology and Climate Research at the Karlsruhe Institute of Technology (KIT) in close cooperation with the Instituto de Astrofísica de Andalucía (CSIC) in Granada, Spain. Results shown in this publication cover both the FR and the RR period.

## 3   Retrieval

The MIPAS Envisat retrieval is based on a non-linear least squares approach and employs a first-order Tikhonov-type regularization (von Clarmann et al., 2003, 2009). The radiative transfer is modelled using the Karlsruhe Optimized and Precise Radiative Transfer Algorithm (KOPRA) model (Stiller, 2000).

The spectral regions used for the retrieval of $CCl_4$ are 772.0 - 791.0 $cm^{-1}$ and 792.0 - 805.0 $cm^{-1}$. The gap from 791.0 to 792.0 $cm^{-1}$ is necessary, since even when accounting for line mixing, strong effects from the $CO_2$ Q-branch still occur in the residuals. Several results from previous steps in the retrieval chain were used to derive $CCl_4$ (Table 1) including the spectral shift ($z_{tangent}$), the temperature (T), the horizontal temperature gradient ($T_{grad}$) and mixing ratio profiles of $HNO_3$, ClO, CFC-11, $C_2H_6$, HCN, $ClONO_2$ and $HNO_4$.

In addition, several species were found to improve the retrieval whenever their mixing ratio profiles were fitted alongside $CCl_4$. These are peroxyacetyl nitrate (PAN), $CH_3CCl_3$, HCFC-22, $O_3$, $H_2O$, $C_2H_2$ and $COF_2$. Although for most of these species results from preceding retrieval steps are avail-

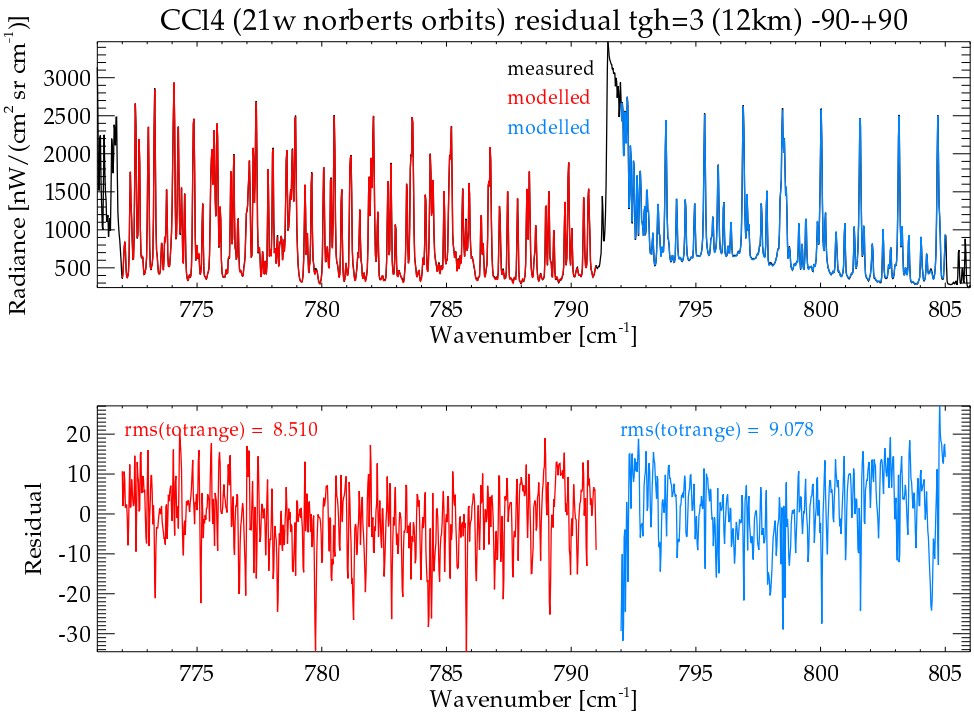

**Figure 1.** Examplary spectra of MIPAS CCl$_4$ at 12 km during the FR period (September 2003). Top panel: spectra; bottom panel: residuals.

able, fitting their concentrations jointly with that of CCl$_4$ reduces the fit residuals significantly. This is attributed to spectroscopic inconsistencies of the interferers' spectroscopic data between the spectral region where these were retrieved and the spectral region where CCl$_4$ is analyzed. Also fitted were a background continuum accounting for spectral contributions from aerosols and a radiance offset which is constant for all tangent altitudes (Table 1).

These retrieval settings lead to spectral fits as displayed in Fig. 1 and Fig. 2, where an example for the FR period and the RR period are shown, respectively. The measured spectra are plotted in black (not discernible from the best fit modelled in the fitting window), while the red and the blue lines represent the modelled spectra of the regions from 772.0 - 791.0 cm$^{-1}$ and 792.0 - 805.0 cm$^{-1}$, respectively. Some periodic residuals are visible in both the FR and the RR period. These result from less than perfectly fitted CO$_2$, but as will be shown in Sec. 5, are only of minor relevance for the accuracy of the retrieved CCl$_4$.

### 3.1 Information cross-talk with PAN

The signature of PAN is particularly prominent in the spectral region of CCl$_4$ and can thus be retrieved during the same retrieval step. Actually, jointly fitting PAN is very important for the CCl$_4$ retrieval. Since PAN was already retrieved from MIPAS spectra before (Glatthor et al., 2007), it is of obvious interest to investigate the PAN results from the CCl$_4$-PAN joint retrieval in comparison with those from the original PAN retrieval. In there CCl$_4$ was fitted alongside PAN but the retrieval was not yet optimized for CCl$_4$.

We find slightly higher volume mixing ratios of PAN throughout most of the altitude-latitude cross-section (Fig. 3). As a consequence, areas showing unphysical mixing ratios below zero in the original retrievals (left panel of Fig. 3) are now slightly positive or very close to zero. This suggests that jointly fit PAN from the retrieval optimized for CCl$_4$ might be more accurate than PAN retrieved using the old CCl$_4$ distributions.

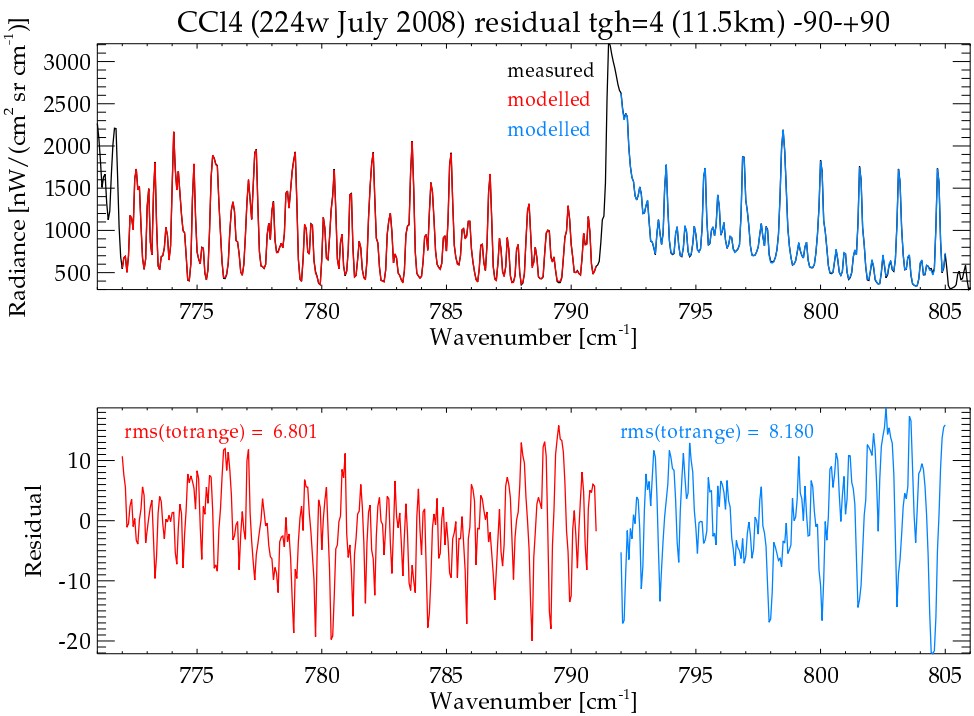

**Figure 2.** Examplary spectra of MIPAS CCl$_4$ at 11.5 km during the RR period (July 2008). Top panel: spectra; bottom panel: residuals.

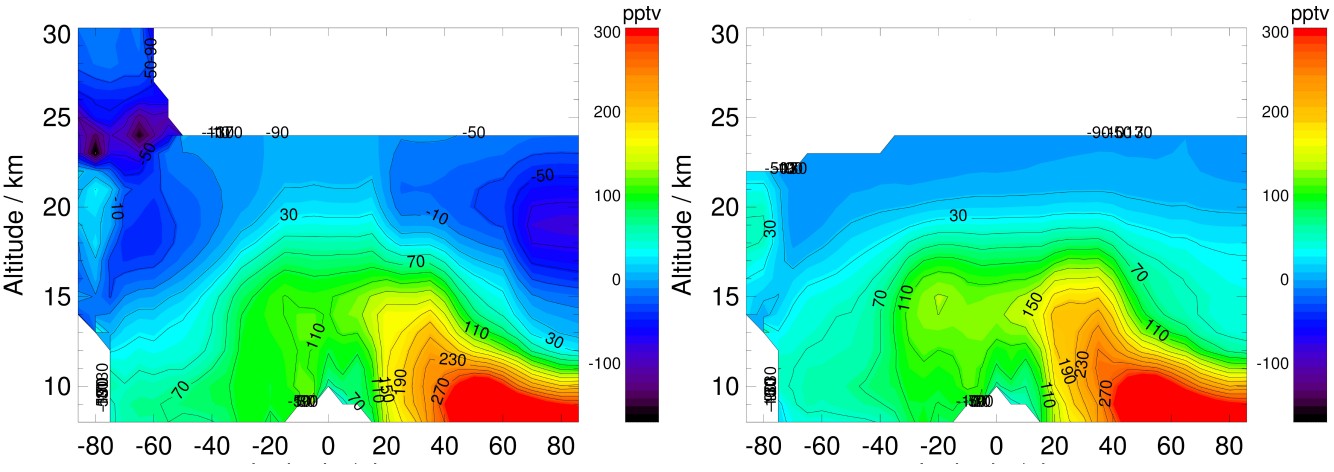

**Figure 3.** PAN altitude/latitude cross-sections (July 2008) from a separate retrieval using the old CCl$_4$ distributions (left) and resulting from a joint retrieval with CCl$_4$ (right).

**Table 1.** Retrieval details on the spectroscopic region, species imported from preceding retrieval steps and variables fitted jointly during the retrieval process. Brackets denote mixing ratios.

| Spectral regions | Imported from preceding retrieval steps | Jointly fitted variables |
| --- | --- | --- |
| 772.0 - 791.0 cm$^{-1}$ | Shift($z_{tangent}$) | [PAN](z) |
| 792.0 - 805.0 cm$^{-1}$ | T(z) | [CH$_3$CCl$_3$](z) |
| | T$_{grad}$(z) | [HCFC-22](z) |
| | [HNO$_3$](z) | [O$_3$](z) |
| | [ClO](z) | [H$_2$O](z) |
| | [CFC-11](z) | [C$_2$H$_2$](z) |
| | [C$_2$H$_6$](z) | [COF$_2$](z) |
| | [HCN](z) | Continuum(z) |
| | [ClONO$_2$](z) | offset |
| | [HNO$_4$](z) | |

## 3.2 Line mixing

Since the spectral region where CCl$_4$ is retrievable contains a CO$_2$ Q-branch, the retrieval is set up to account for line mixing (Funke et al., 1998). This was done by using the Rosenkranz approximation (Rosenkranz, 1975). Tests were also performed using the computationally more demanding direct diagonalisation, but this approach was not found to noticeably change the results of the retrieval. This is possibly the case because the microwindows were carefully selected to omit major spectral signatures of the CO$_2$ Q-branch and because the effect of line mixing is generally smaller at stratospheric pressure levels. However, it was still necessary to omit parts of the CO$_2$ Q-branch. Fig. 4 and Fig. 5 show spectra where the full spectral region was fitted. In Fig. 4, line mixing was not considered and thus a large peak in the residual is visible close to 791.0 cm$^{-1}$. In Fig. 5, the Rosenkranz approximation was used to account for line mixing. Even though the residual is considerably smaller than without line mixing taken into account - as would be expected - peaks significantly larger than for the remainder of the window are still visible between 791.0 and 792.0 cm$^{-1}$. Although inclusion of line mixing significantly reduces the residuals in the CO$_2$ branch, the residuals are still unacceptably large there. With the Rosenkranz approximation, however, the spectral region excluded from the fit could be narrowed to 791.0 to 792.0 cm$^{-1}$ from 790.5 to 792.5 cm$^{-1}$.

## 3.3 New CCl$_4$ Spectroscopic Data

During the ongoing development of the MIPAS Envisat CCl$_4$ retrieval, a new CCl$_4$ spectroscopic dataset was published by Harrison et al. (2017). Fig. 6 shows the influence of these spectroscopic data on an altitude-latitude cross-section of CCl$_4$ distributions of July 2008. The upper panel shows what stratospheric CCl$_4$ distributions retrieved with the original spectroscopic dataset as presented in HITRAN 2000 (Nemtchinov and Varanasi, 2003) look like. The lower panel shows the same cross-section, but using the new spectroscopic dataset by Harrison et al. (2017) for an otherwise identical retrieval setup. While the qualitative and morphological features of the distribution are very similar, lower volume mixing ratios of CCl$_4$ result when the new spectroscopic dataset is used. Comparing these with reported values of ground based measurements as presented in SPARC (2016) indicates that the updated spectroscopic data lead to results which, in the tropopause region, agree better with tropospheric measurements. Tropospheric volume mixing ratios are reported to be at approximately 95 pptv which is very close to what MIPAS Envisat presents around the tropical tropopause and at mid-latitudes of the northern hemisphere when using the new spectroscopic dataset. In contrast, using HITRAN 2000 sometimes results in volume mixing ratios above 100 pptv in the same region. Thus, we consider the new spectroscopic dataset more adequate for the retrieval of CCl$_4$.

## 4 Results

### 4.1 Distributions

Fig. 7, the lower panel of Fig. 6 and Fig. 8 give an overview of the latitudinal and altitude distribution of CCl$_4$ of different time periods. All of the altitude-latitude cross-sections show the expected pattern of CCl$_4$ with a rapid decrease with increasing altitude in the stratosphere, as the gas is photolyzed there. In addition, highest volume mixing ratios appear at the equator where CCl$_4$, along with many other trace gases, enters the stratosphere due to the upward transport associated with the Brewer-Dobson circulation. During January 2010, March 2011 and particularly April 2011, subsidence of higher stratospheric air results in reduced mixing ratios over the North pole. In Spring 2011, an unusually stable northern polar vortex resulted in severe ozone depletion and particularly strong subsidence (Manney et al., 2011; Sinnhuber et al., 2011) which is reflected by the observations shown here. In general, MIPAS Envisat shows higher volume mixing ratios in the lower stratosphere during the FR period, which fits well with the overall decline in CCl$_4$ abundance in the atmosphere due to its restriction under the Montreal Protocol. This impression is also supported by the lower panel in Fig. 6, which shows lower overall volume mixing ratios than MIPAS sees during the FR period, but which are still slightly higher than during 2010 and 2011. All cross-sections show a maximum in the CCl$_4$ volume mixing ratios around the tropical tropopause connected with values of similar magnitude at lower altitudes of northern extratropical regions. This pattern was also seen in HCFC-22 (Chirkov et al., 2016) and

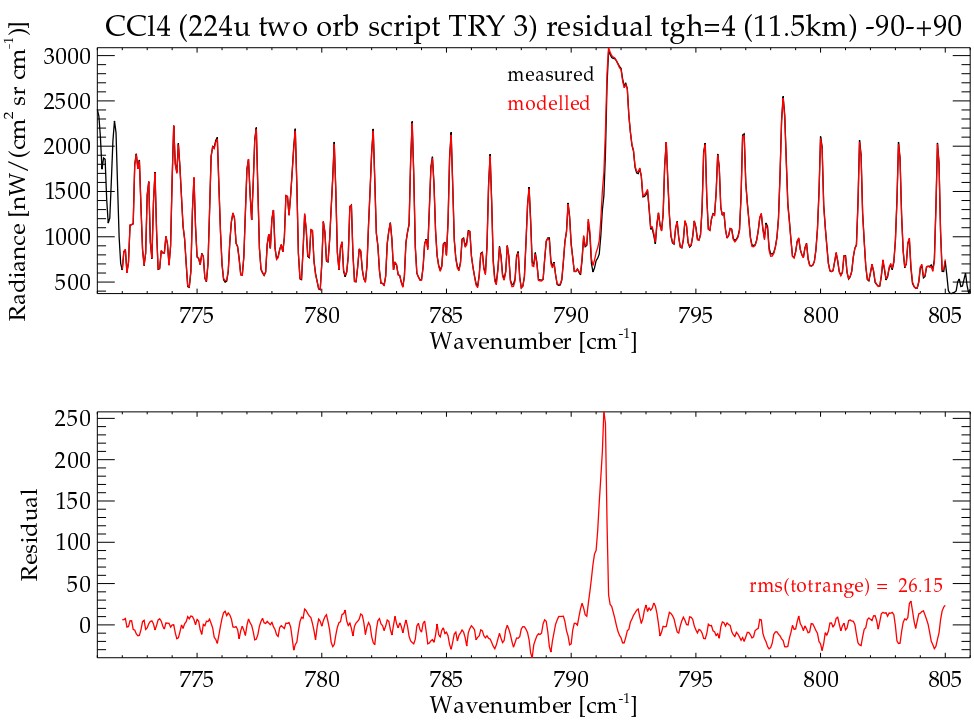

**Figure 4.** Impact of the CO$_2$ Q-branch at 11.5 km altitude without considering line mixing. Top panel: spectra; bottom panel: residuals. Black: measured spectrum, hardly discernible because overplotted by modelled spectra.

could be linked to the Asian monsoon. Calculations with the Chemical Lagrangian Model of the Stratosphere (CLaMS) by Vogel et al. (2016) show that there indeed exists a mechanism which can produce local maxima in the upper troposphere in 2D distributions of source gases. So, the monsoon might offer an explanation for the patterns seen in CCl$_4$ around these atmospheric regions as well.

### 4.2 Altitude Resolution

The vertical resolution of the CCl$_4$ profiles is very similar for the FR and the RR period. From about 2.5-3 km at the lower end of the profiles, it degrades to approximately 5 km at $\sim$ 25 km and $\sim$7 km at $\sim$30 km, calculated as the full width at half maximum of the row of the averaging kernel matrix (Rodgers, 2000). The degrees of freedom are usually around 3.5 for the FR period and close to 4.0 for the RR period (Fig. 9). This is presumably attributed to the finer vertical sampling during the RR period with 27 tangent altitudes compared to 17 tangent altitudes during the FR

period. The signal decreases rapidly with altitude, as the volume mixing ratios of CCl$_4$ do. Above 30 km, hardly any CCl$_4$ information is available in the MIPAS spectra. Slightly below 20 km, the averaging kernels show negative side wiggles which are more pronounced during the FR period (left panel) than the RR period (right panel).

### 4.3 Error Budget

Tables 2 and 3 list the error budgets for mid latitudes during the FR and RR period between 10 and 40 km. Examples for other latitudes can be found in the appendix (Tables A1 and A6). For legibility reasons, the errors are only given every 5 km, although the retrieval grid is 1 km. Errors due to elevation uncertainties of the line of sight and uncertainties of several contributing species are given. All profiles show a strong increase in the relative errors at and above 30 km. During the FR period, the absolute total errors are fairly similar below this altitude, while large differences can occur from 20 km

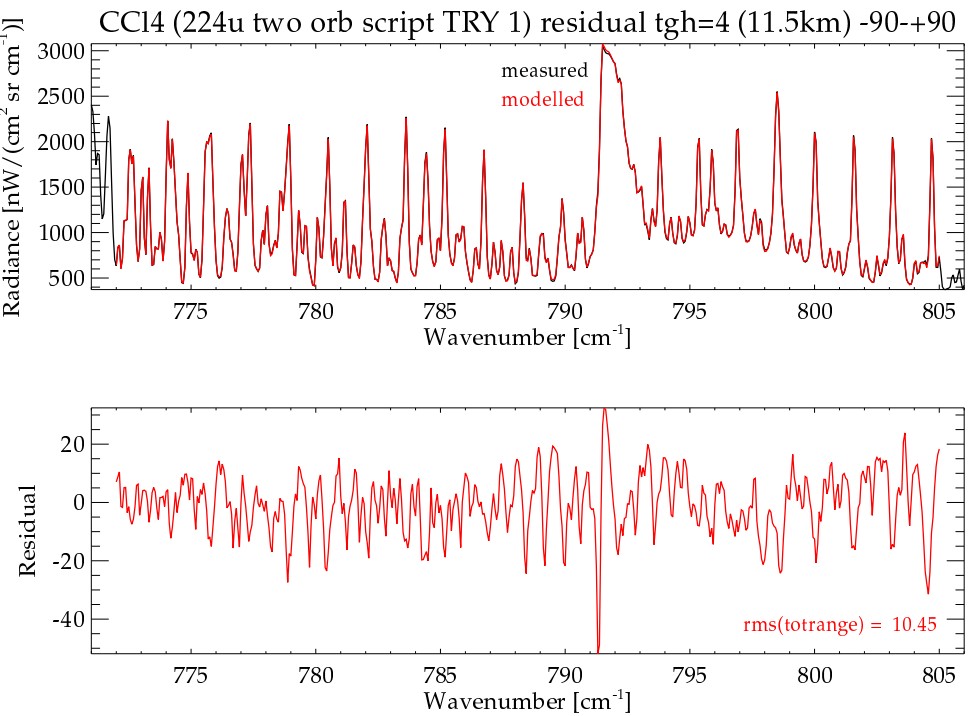

**Figure 5.** Impact of the CO$_2$ Q-branch at 11.5 km altitude taking line mixing it into account (right). Top panel: spectra; bottom panel: residuals. Black: measured spectrum, hardly discernible because overplotted by modelled spectra. Note the different scale of the residual axis compared to Fig. 4.

**Table 2.** Error estimates for a mid-latitude profile during the FR period. Errors are given in pptv (relative errors in %).

| Altitude | total error | noise | total parameter | Gain | LOS | HNO$_4$ | Shift | ILS | Temperature | ClONO$_2$ |
|---|---|---|---|---|---|---|---|---|---|---|
| 40 | 0.0 ( 69.4) | 0.0 ( 57.2) | 0.0 ( 38.8) | 0.0 ( 24.5) | 0.0 ( 22.5) | 0.0 ( 18.2) | 0.0 ( 1.7) | 0.0 ( 9.2) | 0.0 ( 6.3) | 0.0 ( 5.5) |
| 35 | 0.0 ( 68.4) | 0.0 ( 56.7) | 0.0 ( 39.1) | 0.0 ( 23.5) | 0.0 ( 21.5) | 0.0 ( 18.4) | 0.0 ( 1.7) | 0.0 ( 9.0) | 0.0 ( 6.3) | 0.0 ( 5.7) |
| 30 | 0.2 ( 71.0) | 0.2 ( 64.3) | 0.1 ( 33.8) | 0.1 ( 20.3) | 0.1 ( 17.9) | 0.1 ( 20.3) | 0.0 ( 1.8) | 0.0 ( 3.0) | 0.0 ( 5.1) | 0.0 ( 5.1) |
| 25 | 2.3 ( 480.8) | 2.2 ( 459.9) | 0.7 ( 144.2) | 0.4 ( 79.4) | 0.0 ( 3.8) | 0.6 ( 115.0) | 0.0 ( 10.0) | 0.0 ( 0.7) | 0.1 ( 23.0) | 0.1 ( 17.3) |
| 20 | 2.9 ( 5.3) | 2.4 ( 4.4) | 1.6 ( 2.9) | 0.0 ( 0.1) | 1.5 ( 2.8) | 0.1 ( 0.3) | 0.0 ( 0.0) | 0.7 ( 1.2) | 0.1 ( 0.2) | 0.1 ( 0.2) |
| 15 | 5.0 ( 4.9) | 2.1 ( 2.1) | 4.5 ( 4.5) | 0.7 ( 0.7) | 4.0 ( 4.0) | 0.1 ( 0.1) | 0.1 ( 0.1) | 2.0 ( 2.0) | 0.1 ( 0.1) | 0.1 ( 0.1) |
| 10 | 2.7 ( 3.1) | 2.5 ( 2.8) | 0.9 ( 1.0) | 0.2 ( 0.2) | 0.2 ( 0.3) | 0.3 ( 0.3) | 0.1 ( 0.1) | 0.4 ( 0.4) | 0.5 ( 0.6) | 0.1 ( 0.1) |

upwards. Absolute errors are close to 3 pptv between 10 and 25 km, and around 5 to 6 pptv at 15 km where larger volume mixing ratios appear for all atmospheric situations except the polar summer one where the errors stay close to 3 pptv. The largest error component is measurement noise (third column), while at 15 km significant parameter errors have to be considered, in particular the elevation uncertainties of the line of sight (LOS), and instrument line shape (ILS). Beyond this, uncertainties of HNO$_4$ and ClONO$_2$ profiles, frequency calibration (shift) and temperature contribute to the total error. The decrease of retrieval noise towards higher altitudes is explained by the coarser altitude resolution at higher altitudes. For the RR period, the patterns look slightly different. There is no peak in the total error around 15 km, but the total

**Table 3.** Error estimates for a mid-latitude profile during the RR period. Errors are given in pptv (relative errors in %).

| Altitude | total error | noise | total parameter | Gain | LOS | HNO$_4$ | Shift | ILS | Temperature | ClONO$_2$ |
|---|---|---|---|---|---|---|---|---|---|---|
| 40 | 0.0 ( 214.1) | 0.0 ( 127.1) | 0.0 ( 173.9) | 0.0 ( 73.6) | 0.0 ( 147.2) | 0.0 ( 24.8) | 0.0 ( 2.5) | 0.0 ( 24.8) | 0.0 ( 24.1) | 0.0 ( 13.4) |
| 35 | 0.0 ( 211.3) | 0.0 ( 128.1) | 0.0 ( 172.9) | 0.0 ( 70.4) | 0.0 ( 147.3) | 0.0 ( 25.0) | 0.0 ( 2.6) | 0.0 ( 24.3) | 0.0 ( 23.7) | 0.0 ( 13.4) |
| 30 | 0.2 ( 141.2) | 0.1 ( 123.6) | 0.1 ( 61.8) | 0.0 ( 15.9) | 0.1 ( 47.7) | 0.0 ( 24.7) | 0.0 ( 2.8) | 0.0 ( 22.1) | 0.0 ( 2.8) | 0.0 ( 11.5) |
| 25 | 2.4 ( 187.3) | 2.2 ( 171.7) | 0.9 ( 67.1) | 0.2 ( 14.0) | 0.4 ( 30.4) | 0.4 ( 33.6) | 0.1 ( 4.8) | 0.6 ( 44.5) | 0.0 ( 0.0) | 0.2 ( 16.4) |
| 20 | 3.5 ( 15.0) | 2.6 ( 11.1) | 2.4 ( 10.3) | 0.1 ( 0.4) | 2.3 ( 9.9) | 0.1 ( 0.4) | 0.1 ( 0.3) | 0.1 ( 0.5) | 0.1 ( 0.2) | 0.0 ( 0.1) |
| 15 | 3.3 ( 6.1) | 2.0 ( 3.7) | 2.6 ( 4.8) | 0.5 ( 1.0) | 2.5 ( 4.6) | 0.1 ( 0.3) | 0.0 ( 0.1) | 0.1 ( 0.2) | 0.1 ( 0.1) | 0.0 ( 0.0) |
| 10 | 5.7 ( 6.1) | 4.3 ( 4.6) | 3.7 ( 4.0) | 1.1 ( 1.2) | 3.5 ( 3.8) | 0.2 ( 0.2) | 0.0 ( 0.0) | 0.4 ( 0.4) | 0.4 ( 0.4) | 0.1 ( 0.1) |

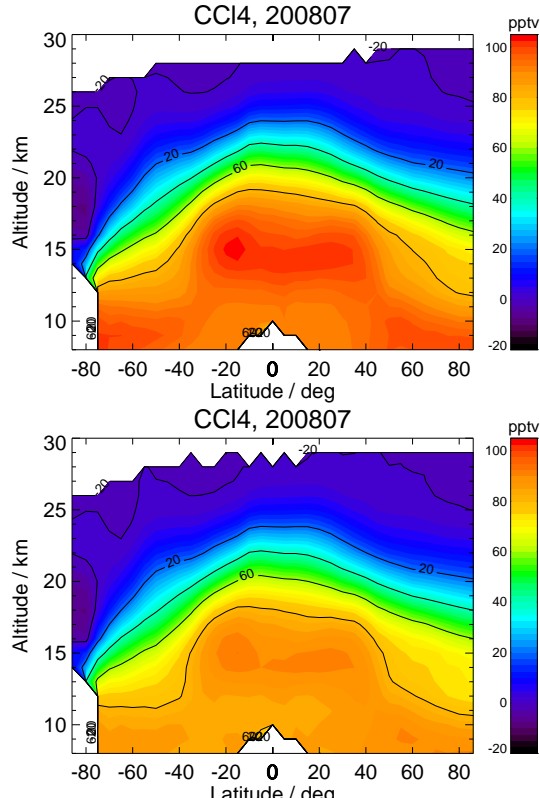

**Figure 6.** Altitude-latitude cross-section of July 2008, using the spectroscopic dataset by Nemtchinov and Varanasi (2003) (top) and using the new spectroscopic data by Harrison et al. (2017).

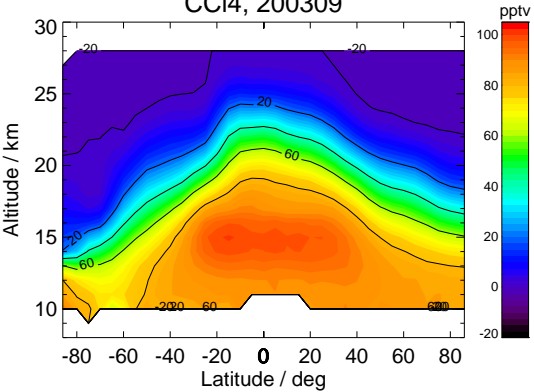

**Figure 7.** Altitude-latitude cross-sections of MIPAS CCl$_4$ for the FR period (September 2003).

proximately 18 km. Correspondingly, the error estimate can be considered realistic from the bottom of the profile up to this altitude.

### 4.4 Trends

Fig. 11 shows an altitude-latitude cross-section of MIPAS Envisat CCl$_4$ trends. These trends were estimated by the same method as described by Eckert et al. (2014), which is based on the method by von Clarmann et al. (2010). In addition to the setup used by Eckert et al. (2014), the El-Niño-Southern Oscillation (ENSO) was also taken into account. The data set used for trend calculation covers the entire MIPAS Envisat measurement period from July 2002 to April 2012. The distribution of the trends agrees well with the trends estimated by Valeri et al. (2017), who calculated trends from MIPAS Envisat V7 data they formerly retrieved and displayed them on a pressure-latitude grid. The most likely cause of differences between their and our trend estimates are the underlying MIPAS spectra. We use MIPAS V5 spectra which were found to be subject to an instrument drift due to detector aging (Eckert et al., 2014). Valeri et al. (2017) use version 7 spectra where an attempt was made to tackle the problem of detector aging during the level-1

error is either rather constant at lower altitudes or decreases with altitude. Contributions to the error budget are, however, similar to the FR period.

Fig. 10 compares the estimated total error with the deviation of the profiles in a quiescent atmosphere. This comparison was created in a similar way as in Eckert et al. (2016, Sec. 6). Up to 18 km altitude, the sample standard deviation of MIPAS Envisat results is only slightly larger than the estimated error. Thus, these profiles suggest that the estimated error can explain most of the variability in the CCl$_4$ profiles up to ap-

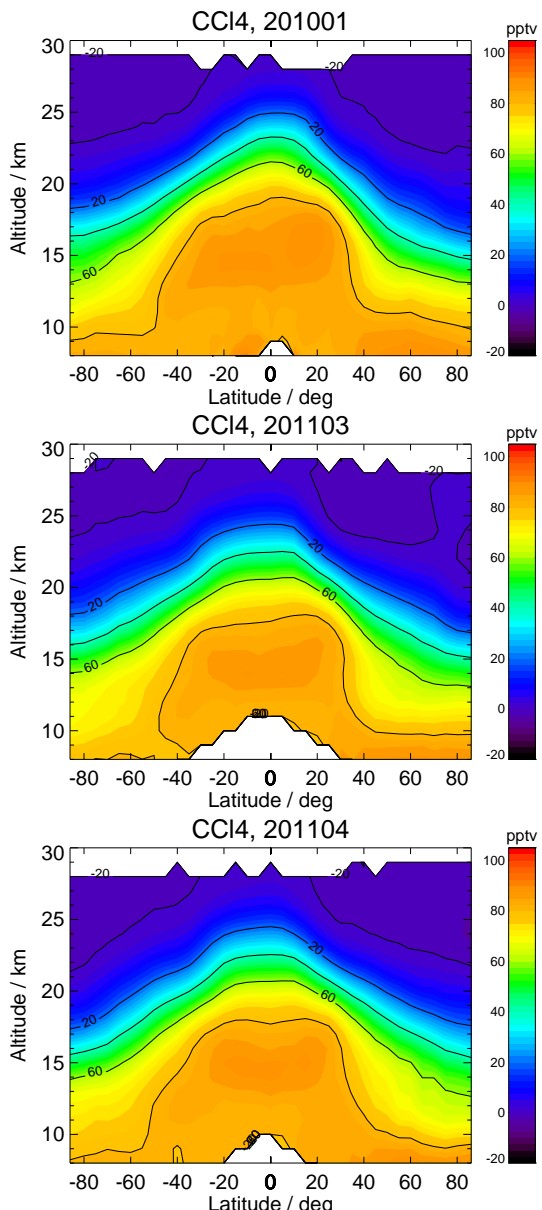

**Figure 8.** Altitude-latitude cross-sections of MIPAS CCl$_4$ for the RR period. Top to bottom: July 2008, January 2010 and March and April 2011.

processing. However, Hubert et al. (2016) show that there is still a drift problem in the version 7 MIPAS temperatures. Since these temperature drifts are expected to propagate onto the retrieved CCl$_4$ mixing ratios, it is not clear if version 5 or version 7 is more adequate for trend analysis. In spite of these differences and technical differences in the level-2 data processing, the trends inferred by Valeri et al. (2017) and ours show important common features. In both data sets a hemispheric asymmetry is found, with positive trends in

the Southern hemisphere around 25 km (however, the region is larger in our data set) and negative trends in the Northern hemisphere in almost the whole altitude range. Also the average magnitudes of the inferred trends agree reasonably well between both data sets.

## 5 Comparisons

### 5.1 Historical comparisons

#### 5.1.1 ATMOS

The ATMOS (Atmospheric Trace Molecule Spectroscopy) instrument measured in solar occultation covering the spectral region from 600 to 4700 cm$^{-1}$ with a spectral resolution of 0.01 cm$^{-1}$. ATMOS took measurements in 1985, 1992, 1993 and 1994. The ATMOS profiles shown in Fig. 12 were extracted directly from Zander et al. (1996, Fig. 1). CCl$_4$ volume mixing ratio profiles in the subtropics (20-35°N; thin dashed lines) and at midlatitudes (35-49°N; thin full lines) are presented there. Measurements were taken from 3 to 12 November in 1994 during the ATLAS-3 shuttle mission. We adopted depicting midlatitude profiles as solid lines and subtropic profiles as dashed lines in Fig. 12 of this manuscript. To compare the ATMOS profiles with MIPAS Envisat, we used MIPAS Envisat data of all years from 3 to 12 November and calculated an arithmetic mean for both latitude bands (subtropics and midlatitudes). In Fig. 12, MIPAS Envisat profiles are shown in blue, while the ATMOS profiles are shown in orange. The ATMOS profiles show higher volume mixing ratios than those of MIPAS Envisat, because they were measured shortly after CCl$_4$ emissions were restricted and, thus, volume mixing ratios used to be higher in the atmosphere. However, the general shapes of the ATMOS profiles agree well with those of MIPAS Envisat. Both, MIPAS Envisat and ATMOS, show CCl$_4$ mixing ratios which quickly decrease with altitude. The slopes of decline are similar above ~20 km. Largest differences are visible at the lower end of the midlatitude profiles. ATMOS CCl$_4$ mixing ratios also agree well with Liang et al. (2016, Fig. 2) where a time series of CCl$_4$ surface mixing ratios over several decades is shown. Volume mixing ratios at the lower end of the profiles are noticeably higher than 100 pptv, which is in very good agreement with peak values of CCl$_4$ shown in Liang et al. (2016, Fig. 2) for the time around and shortly after 1990. Taking the temporal development of the surface mixing ratios into account, ATMOS and MIPAS Envisat measurements provide a coherent picture.

#### 5.1.2 MIPAS-B

The first balloon-borne version of the MIPAS instrument was developed prior to the satellite instrument in the late 1980's and early 1990's at the Institute of Meteorology and Climate Research (IMK) in Karlsruhe (Fischer and Oelhaf,

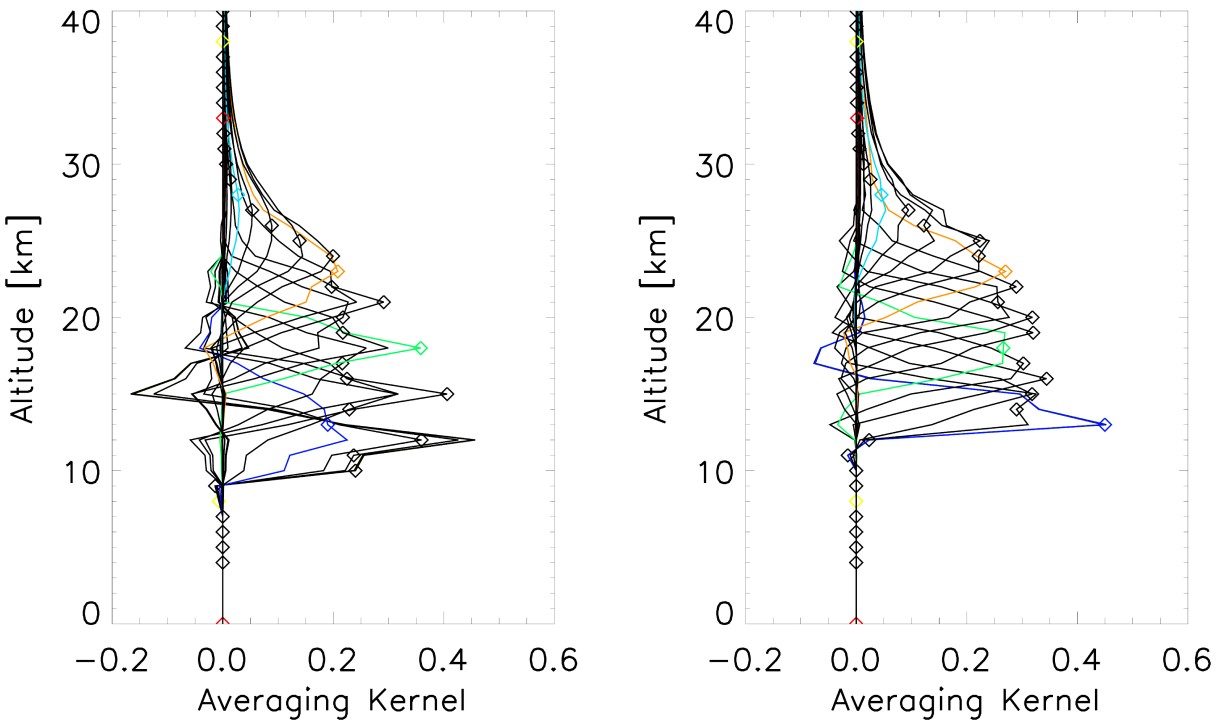

**Figure 9.** Rows of exemplary Averaging Kernels of MIPAS CCl$_4$. Left: FR period (September 2003). Right: RR period (July 2008).

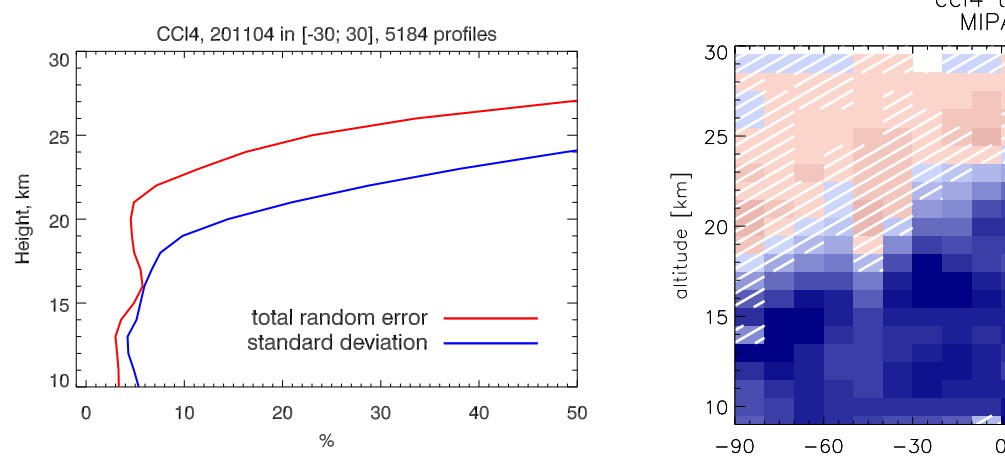

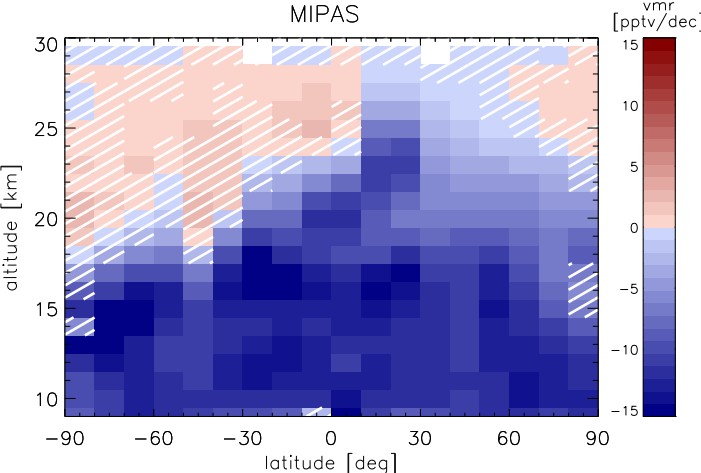

**Figure 10.** Comparison of the estimated total error with the standard deviation of several MIPAS profiles for a quiescent atmospheric situation (equator). Red: total error budget, blue: standard deviation.

**Figure 11.** Altitude-latitude cross-sections of MIPAS CCl$_4$ trends covering the entire measurement period from July 2002 to April 2012. Red colours indicate increasing CCl$_4$ volume mixing ratios. Blue colours indicate declining CCl$_4$ concentrations. Hatching shows where no statistically significant trends could be calculated at two sigma confidence level.

1996). Measurements with this instrument have been taken since 1989 (von Clarmann et al., 1993) and first profiles of CCl$_4$ were derived from a flight at Kiruna, Sweden, on 14 March 1992 (von Clarmann et al., 1995). Due to the strong

decrease of CCl$_4$ with altitude, a clear signal of the gas could

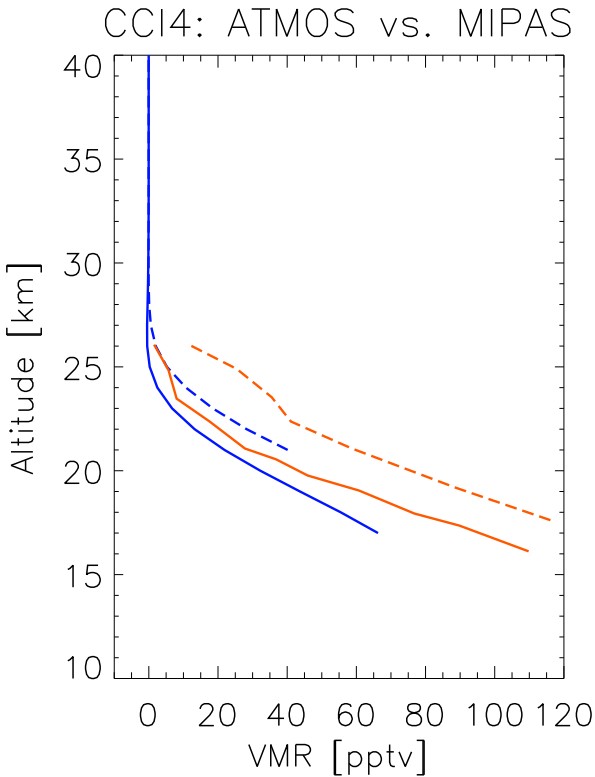

**Figure 12.** Qualitative comparison of profiles from ATMOS (orange) taken during the ATLAS-3 mission (as shown in Zander et al. (1996, Fig. 1)) and climatological means of MIPAS (blue) during 3-12 November of each year. Solid lines refer to midlatitude measurements (35-49°N). Dashed lines indicate subtropical measurements (20-35°N).

not be identified at tangent altitudes of 14.5 km and above. Thus, only the spectrum at 11.3 km was analyzed and the total amount of CCl$_4$ was estimated by scaling the vertical profile and using information on the shape as measured in polar winter conditions before. This leads to an estimated concentration of approximately 110 pptv at 11.3 km, which is slightly higher than the peak surface values in the long time series of CCl$_4$ shown in Liang et al. (2016). Ground based measurements shown in there support favouring the MIPAS Envisat CCl$_4$ retrieval with the new spectroscopic dataset, since respective results agree better with measurements shown in Liang et al. (2016). MIPAS-B results overestimate the ground based measurements slightly providing a consistent picture when taking differences in the volume mixing ratios into account which result from the old versus the new spectroscopic dataset.

## 5.2 Comparisons with collocated measurements

All collocated measurements were analyzed using spectroscopic data of Nemtchinov and Varanasi (2003), which are included in the HITRAN 2000 database (Rothman et al., 2003). Thus, in order to allow for a meaningful comparison and not to mask possible other differences, a dedicated MIPAS Envisat comparison dataset was generated which is based on these spectroscopic data as well.

### 5.2.1 ACE-FTS

The Atmospheric Chemistry Experiment Fourier Transform Spectrometer ACE-FTS is one of two instruments aboard the Canadian Satellite SCISAT-1. On 12 August 2003, it was launched into a 74° orbit at 650 km to ensure a focus on higher latitudes. It covers the globe from 85°S to 85°N. Since ACE-FTS is an occultation instrument, it takes measurements during 15 sunrises and 15 sunsets a day within two latitude bands. The vertical scan range covers altitudes from the middle troposphere up to 150 km. Wavelengths between 750 cm$^{-1}$ and 4400 cm$^{-1}$ (13.3 μm and 2.3 μm) can be detected with a spectral resolution of 0.02 cm$^{-1}$. The vertical sampling depends on the altitude as well as the beta angle. The latter is the angle between the orbit track and the path from the instrument to the sun. The sampling ranges from ~1 km between 10 km and 20 km to ~2-3.5 km around 35 km and declines to 5-6 km at the upper end of the vertical range. The field of view covers 3-4 km, which is approximately similar to the vertical resolution of the instrument. Comparisons in this study were made using version 3.5 of the ACE-FTS data. The CCl$_4$ retrieval is performed between 787.5 cm$^{-1}$ and 805.5 cm$^{-1}$ at altitudes from 7 km to 25 km (Allen et al., 2009).

For the comparison with ACE-FTS (Fig. 13), coincident profiles within 2 hours time difference and no further than 5° latitude and 10° longitude away were used. Profiles at latitudes higher than 60°S were omitted. Between the lower end and ~16 km the agreement is always close to 10 %, with slightly larger differences below 10 km than between 10 and 15 km. Above 15 km, the mean profiles deviate more strongly and exceed relative differences of 50 % above 19 km (Fig. 13d)). However, differences above 19 km are not as apparent in the absolute comparison (Fig. 13a)). The volume mixing ratio difference stays within similar values up to near 25 km. Since CCl$_4$ decreases rapidly with altitude, this difference is far more pronounced in relative terms. MIPAS shows slightly lower volume mixing ratios than ACE-FTS, in general. Part of this might be attributed to PAN not being accounted for in the ACE-FTS v3.5 retrieval (Harrison et al., 2017). With PAN missing from the forward model calculations, the retrieval increases CCl$_4$ to compensate. Preliminary ACE-FTS version 4 results indicate that retrieved CCl$_4$ will skew lower when PAN is included. However, Harrison et al. (2017) do not investigate the magnitude of the

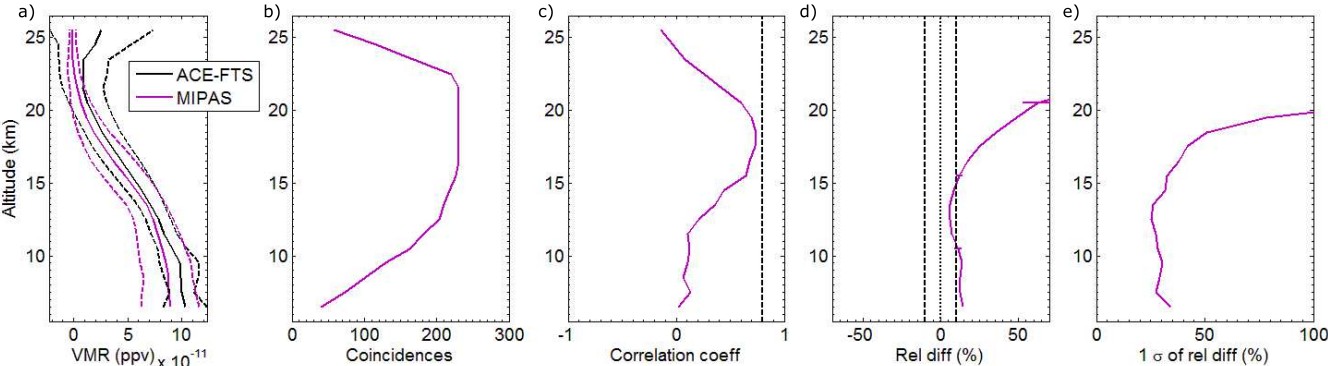

**Figure 13.** Comparison of MIPAS Envisat and version 3.5 ACE-FTS CCl$_4$. a) Mean profiles of all coincident profiles (black: ACE-FTS, magenta: MIPAS). Dashed lines show the standard deviations of the mean profiles. b) Number of coincident points per altitude. c) Correlation coefficient of the mean profiles. d) Relative differences of the mean profiles. e) One standard deviation of the relative differences of the mean profiles.

effect of including PAN versus not including it. Other items changed in the retrieval e.g. the microwindow set and new cross sections, so it is not clear how much of the decrease in CCl$_4$ can be attributed to the inclusion of PAN as an interferer in the ACE-FTS retrieval. Nevertheless, the agreement between MIPAS Envisat and ACE is very good, staying within the 10 % range for the differences up to above 15 km.

### 5.2.2 MIPAS-B2

MIPAS-B2 is the follow-up of MIPAS-B (Friedl-Vallon et al., 2004) which was lost in 1992. MIPAS-B and MIPAS-B2 measurements add up to more than 20 flights to date. MIPAS-B2 covers the spectral range from $750\,\mathrm{cm}^{-1}$ to $2500\,\mathrm{cm}^{-1}$ ($13.3\,\mu\mathrm{m}$ and $4\,\mu\mathrm{m}$) and vertical ranges up to the floating altitude of typically around 30-40 km. The vertical sampling is approximately 1.5 km. The spectral region used for the MIPAS-B2 retrieval ranges from 786.0 to $806.0\,\mathrm{cm}^{-1}$. MIPAS-B2 and MIPAS Envisat use the same retrieval strategy and forward model to derive vertical profiles.

Fig. 14 and Fig. 15 show CCl$_4$ measurements from a single flight of MIPAS-B2 each, compared with collocated measurements of MIPAS Envisat along diabatic 2-day backward and forward trajectories. These trajectories were calculated at Free University of Berlin (Naujokat and Grunow, 2003) and are based on European Centre for Medium-Range Weather Forecasts (ECMWF) $1.25°$x $1.25°$analyses. The trajectories start at different altitudes at the respective geolocation of the balloon measurement. Coincidence criteria for this comparison were 1 h and 500 km within the temporal and spacial range of the balloon location. Fig. 14 shows a comparison with the MIPAS-B2 flight on 24 January 2010. The comparison with the MIPAS Envisat mean profile (red line), which was calculated from the

ensemble of all collocated MIPAS Envisat measurements (red squares), agrees with the MIPAS-B2 measurement (black line) within 5 pptv for most of the altitude range. The MIPAS-B2 measurement lies well within the spread of all collocated MIPAS Envisat profiles. The difference (middle panel) is always close to the total combined error, which includes all error estimates except the spectroscopy error. The latter has not been included because a MIPAS Envisat retrieval setup was used for this comparison which is based on the same spectroscopic data as the MIPAS-B2 retrieval. The right panel shows the relative error, which stays well within 5 % up to 17 km. Only between 16 and 18 km, the relative difference noticeably exceeds the combined error of the instruments.

The comparison of the MIPAS-B2 flight on 31 March 2011 (Fig. 15) with MIPAS Envisat presents even better agreement. The difference between the two profiles never exceeds 5 pptv (middle panel) and stays within or close to the combined error of the instruments throughout the whole altitude range. Larger deviations in the relative differences only occur above 18 km, where the combined error of the instruments also increases rapidly, because of small volume mixing ratios of CCl$_4$. Overall, the comparisons with MIPAS-B2 show excellent agreement between the two instruments. This suggests that the MIPAS Envisat CCl$_4$ error estimate are realistic and that the residuals in the CO$_2$ lines mentioned in Sec. 3.2 have no major impact on the CCl$_4$ retrieval. This is also supported by Fig. 10, at least up to about 18 km, since the standard deviation of the profiles can be explained by the MIPAS Envisat error estimates to a large extent.

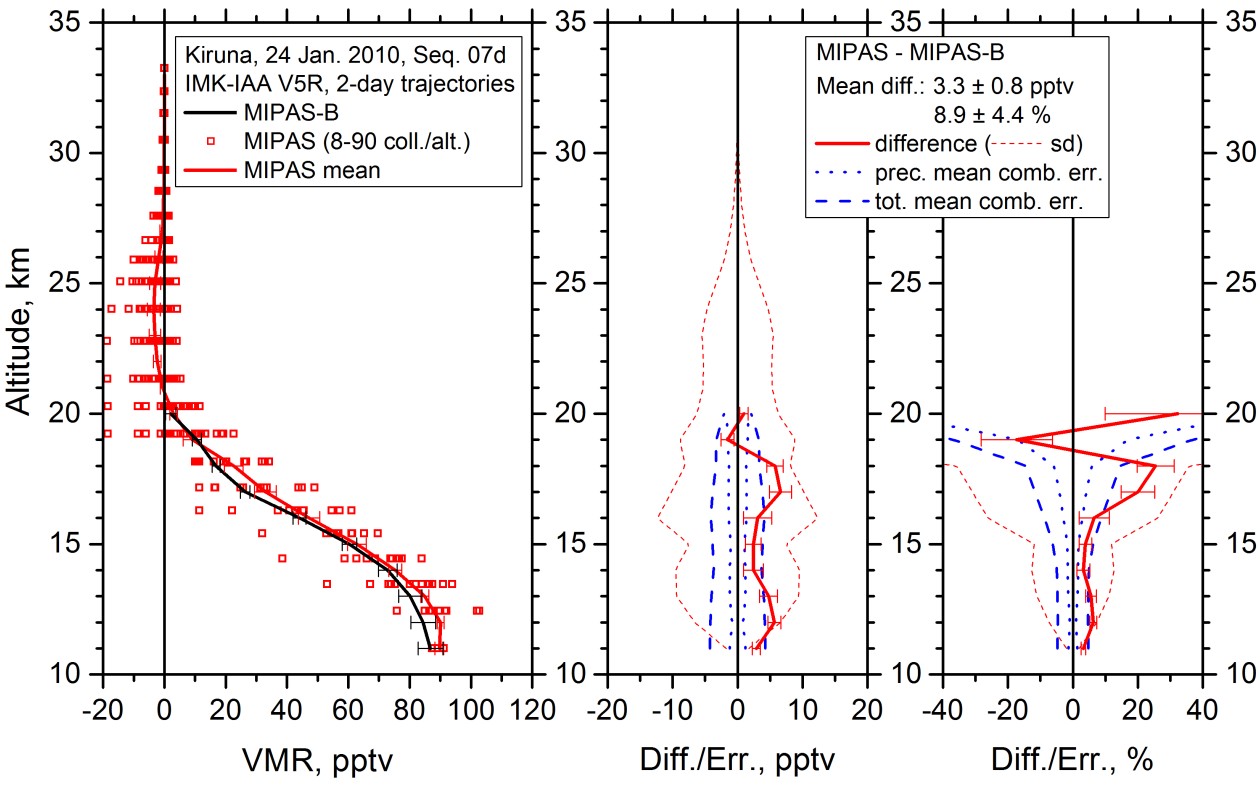

**Figure 14.** Comparison of MIPAS Envisat and MIPAS-B2 CCl$_4$ for the MIPAS-B2 flight on 24 January 2010 over Kiruna, Sweden. Left: Mean profile of all coincident profiles (black line: MIPAS-B2, red line: MIPAS mean, red squares: coincident MIPAS measurements). Middle: absolute total error budget without consideration of the spectroscopy error. Right: relative error budget - red continuous line: difference between the mean profiles; red dotted line: standard deviation; blue dotted line: mean combined precision; blue dashed line: total mean combined error.

### 5.2.3 Cryosampler

The Cryosampler whose measurements are used here was developed at Forschungszentrum Jülich (Germany) in the early 1980s (Schmidt et al., 1987) and is a balloon-borne instrument. It collects whole air samples which are then frozen during the flight and analyzed using gas chromatography after the flight. In this analysis, a flight performed on 1 April 2011 by University of Frankfurt (Fig. 16 black circles) is compared to collocated MIPAS Envisat profiles that lie within 1000 km and 24 h of the Cryosampler profile. The MIPAS Envisat profiles used for the comparison are those retrieved with the new spectroscopic dataset (continuous blue line: closest MIPAS profile, red line: MIPAS mean profile, blue-greyish lines: all collocated MIPAS profiles). In addition, the closest profile produced with the old spectroscopic dataset is shown (dashed blue line). The only difference between the blue line and the

dashed blue line are the different spectroscopic datasets. It is clearly visible that the closest MIPAS profile produced with the new spectroscopic data comes closer to the Cryosampler measurements, even though these still show slightly lower volume mixing ratios of CCl$_4$. A similar pattern of two outliers (second and forth lowest Cryosampler measurements) were also seen in a comparison of Cryosampler and MIPAS measurements of CFC-11 and CFC-12 (Eckert et al., 2016), even though the second lowest outlier is not as obvious for the CFCs. However, this might be an indication that Cryosampler captured fine structures (like laminae) produced by the unique atmospheric situation in spring 2011 (Manney et al., 2011; Sinnhuber et al., 2011), that MIPAS Envisat cannot resolve due to its coarser vertical resolution. All other Cryosampler measurements lie within the spread of the collocated MIPAS Envisat profiles. Taking this into account, the overall agreement of MIPAS and Cryosampler

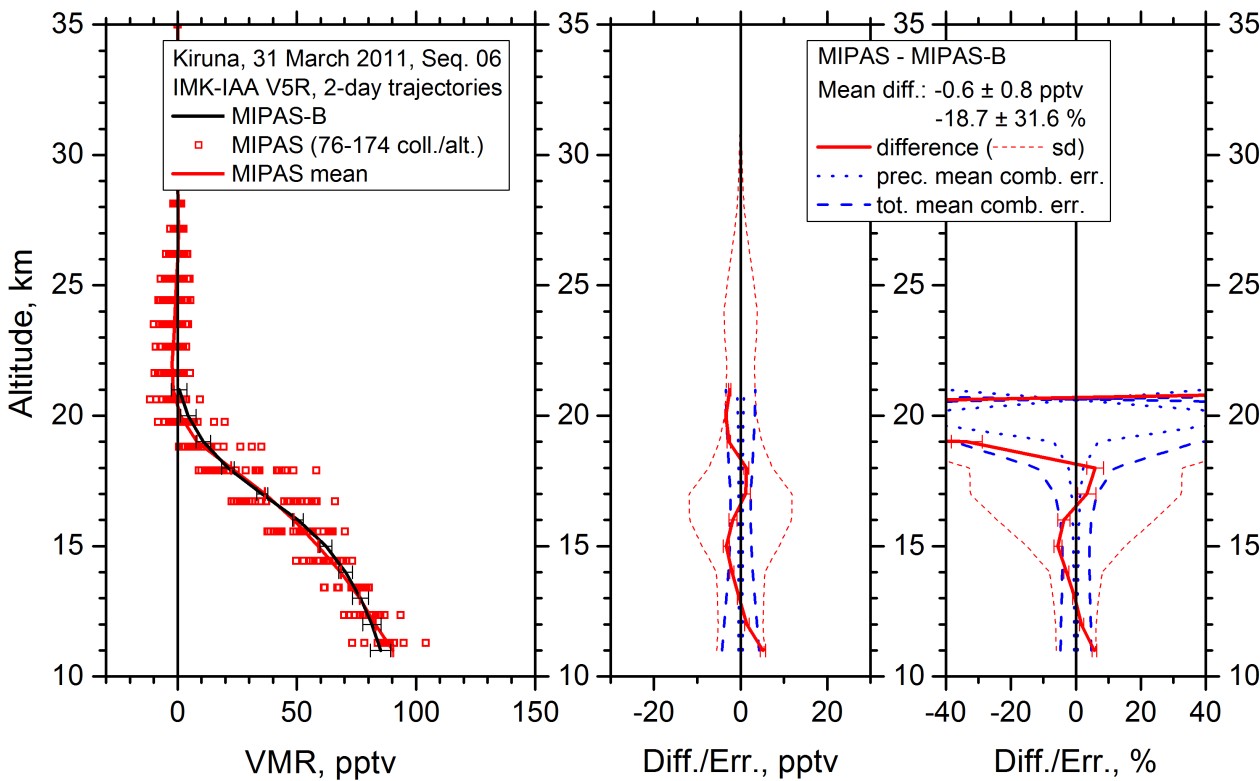

**Figure 15.** Comparison of MIPAS Envisat and MIPAS-B2 $CCl_4$ for the MIPAS-B2 flight on 31 March 2011 over Kiruna, Sweden. Left: Mean profile of all coincident profiles (black line: MIPAS-B2, red line: MIPAS mean, red squares: coincident MIPAS measurements). Middle: absolute total error budget without consideration of the spectroscopy error. Right: relative error budget - red continuous line: difference between the mean profiles; red dotted line: standard deviation; blue dotted line: mean combined precision; blue dashed line: total mean combined error.

is good and Fig. 16 supports the assumption that the retrieval is improved by the usage of the new spectroscopic dataset.

## 6   Conclusions

Vertical profiles of $CCl_4$ were retrieved from MIPAS Envisat limb emission spectra considering various interfering trace gases and with PAN playing a particularly important role. Using line-mixing in the forward model made it possible to narrow the spectral region that had to be omitted due to large residuals and thus to include additional information useful for the retrieval of $CCl_4$, even though parts of the $CO_2$ Q-branch had still to be excluded. Introducing a new spectroscopic dataset (Harrison et al., 2017) resulted in lower volume mixing ratios of $CCl_4$ which agree better with other results, e.g. tropospheric values shown in Liang et al. (2016) and Cryosampler measurements. The expected atmospheric

distribution patterns are clearly visible in altitude-latitude cross-sections. These show higher volume mixing ratios of $CCl_4$ in the tropics and at lower altitudes which quickly decrease above the tropopause due to photolyzation. They also decrease with increasing latitude and thus follow the Brewer-Dobson circulation. A maximum in the tropics connected with higher values of $CCl_4$ below the northern extra-tropical tropopause is a feature also seen in HCFC-22 (Chirkov et al., 2016) where it was associated with the uplift in the Asian monsoon, so $CCl_4$ distributions in this region might have a similar explanation. Trends of the entire measurement period from July 2002 to April 2012 show good agreement with trends estimated by Valeri et al. (2017). Comparisons with ACE-FTS and MIPAS-B2 show very good agreement and historical measurements of MIPAS-B2 and ATMOS are coherent with MIPAS Envisat $CCl_4$ results using the new spectroscopic data. MIPAS profiles retrieved using the new spectroscopic dataset agree well with Cryosampler and deviations

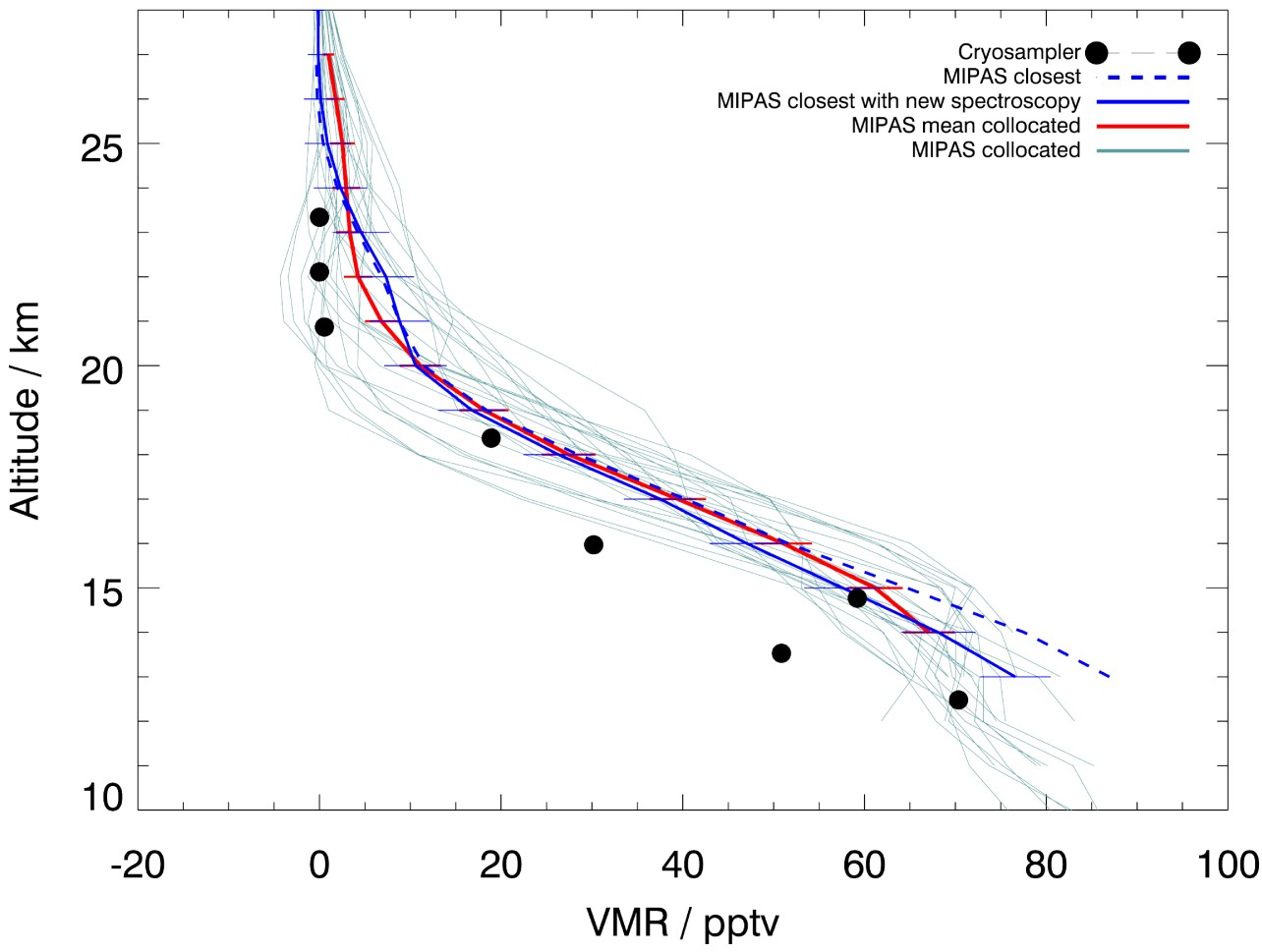

**Figure 16.** Comparison of MIPAS Envisat and cryosampler CCl$_4$. The cryosampler measurement taken on 1 April 2011. The continuous and dashed blue lines are the respective closest MIPAS Envisat profiles with the new and the old spectroscopic dataset.

between the measurements can be explained reasonably. The latter comparison also suggests that the new spectroscopic dataset improves the MIPAS Envisat CCl$_4$ retrieval. The MIPAS Envisat estimated error can explain most of the variability of a set of profiles measured during quiescent atmospheric conditions up to 18 km, so the error estimate seems to be realistic. This is also supported by the comparison of MIPAS Envisat and MIPAS-B2 where the differences between the measurements stay mostly within the combined error of the instruments. Putting differences resulting from different special resolutions aside, also the comparison with the Cryosampler profile suggests to favour the spectroscopic dataset introduced by Harrison et al. (2017) over the dataset used before.

*Acknowledgements.* The retrievals of IMK/IAA were partly performed on the HP XC4000 of the Scientific Supercomputing Center (SSC) Karlsruhe under project grant MIPAS. IMK data analysis was supported by DLR under contract number 50EE0901. MIPAS level 1B data were provided by ESA. We acknowledge support by Deutsche Forschungsgemeinschaft and Open Access Publishing Fund of Karlsruhe Institute of Technology. This work was supported by the DFG project for the 'Consideration of lifetimes of tracers for the determination of stratospheric age spectra and the Brewer-Dobson Circulation (COLIBRI)'. The Atmospheric Chemistry Experiment (ACE), also known as SCISAT, is a Canadian-led mission mainly supported by the Canadian Space Agency and the Natural Sciences and Engineering Research Council of Canada. Balloon flights and data analysis of MIPAS-B data used here were supported by the European Space Agency (ESA), the German Aerospace Center (DLR), CNRS (Centre National de la Recherche Scientifique), and CNES (Centre National d'Etudes Spatiales).

**Appendix A:  Error Estimates**

**Table A1.** Error estimate for an equatorial profile during the FR period. Errors are given in pptv (relative errors in %).

| Altitude | total error | noise | total parameter | Gain | LOS | HNO$_4$ | Shift | ILS | Temperature | ClONO$_2$ |
|---|---|---|---|---|---|---|---|---|---|---|
| 40 | 0.0 ( 210.6) | 0.0 ( 178.7) | 0.0 ( 114.8) | 0.0 ( 70.2) | 0.0 ( 45.3) | 0.0 ( 55.5) | 0.0 ( 6.0) | 0.0 ( 37.6) | 0.0 ( 30.0) | 0.0 ( 17.2) |
| 35 | 0.0 ( 214.1) | 0.0 ( 183.5) | 0.0 ( 116.2) | 0.0 ( 67.3) | 0.0 ( 45.3) | 0.0 ( 55.7) | 0.0 ( 6.0) | 0.0 ( 37.3) | 0.0 ( 30.0) | 0.0 ( 17.1) |
| 30 | 0.2 ( 195.8) | 0.2 ( 177.1) | 0.1 ( 85.8) | 0.1 ( 51.3) | 0.0 ( 23.3) | 0.1 ( 54.1) | 0.0 ( 5.2) | 0.0 ( 17.7) | 0.0 ( 23.3) | 0.0 ( 14.0) |
| 25 | 2.3 ( 30.4) | 2.2 ( 29.0) | 0.9 ( 11.9) | 0.4 ( 4.8) | 0.5 ( 7.1) | 0.5 ( 7.1) | 0.1 ( 0.8) | 0.2 ( 2.6) | 0.2 ( 2.8) | 0.1 ( 1.3) |
| 20 | 2.8 ( 3.8) | 2.5 ( 3.4) | 1.3 ( 1.8) | 0.2 ( 0.2) | 0.8 ( 1.2) | 0.1 ( 0.2) | 0.0 ( 0.0) | 0.9 ( 1.2) | 0.3 ( 0.4) | 0.1 ( 0.2) |
| 15 | 5.3 ( 5.5) | 2.2 ( 2.3) | 4.9 ( 5.1) | 0.9 ( 1.0) | 4.2 ( 4.4) | 0.2 ( 0.2) | 0.1 ( 0.1) | 2.3 ( 2.4) | 0.4 ( 0.4) | 0.1 ( 0.1) |
| 10 | 2.8 ( 3.2) | 2.6 ( 2.9) | 1.0 ( 1.1) | 0.2 ( 0.2) | 0.1 ( 0.1) | 0.2 ( 0.2) | 0.1 ( 0.1) | 0.3 ( 0.4) | 0.8 ( 0.9) | 0.1 ( 0.1) |

**Table A2.** Error estimate for a polar summer profile during the FR period. Errors are given in pptv (relative errors in %).

| Altitude | total error | noise | total parameter | Gain | LOS | HNO$_4$ | Shift | ILS | Temperature | ClONO$_2$ |
|---|---|---|---|---|---|---|---|---|---|---|
| 40 | 0.0 ( 95.1) | 0.0 ( 64.2) | 0.0 ( 69.4) | 0.0 ( 38.5) | 0.0 ( 46.2) | 0.0 ( 19.8) | 0.0 ( 1.4) | 0.0 ( 19.0) | 0.0 ( 11.3) | 0.0 ( 5.1) |
| 35 | 0.0 ( 93.7) | 0.0 ( 64.1) | 0.0 ( 69.0) | 0.0 ( 39.4) | 0.0 ( 46.8) | 0.0 ( 19.7) | 0.0 ( 1.4) | 0.0 ( 19.0) | 0.0 ( 11.3) | 0.0 ( 5.2) |
| 30 | 0.2 ( 117.2) | 0.2 ( 87.9) | 0.1 ( 73.2) | 0.1 ( 39.5) | 0.1 ( 53.7) | 0.1 ( 26.4) | 0.0 ( 1.8) | 0.0 ( 11.2) | 0.0 ( 11.2) | 0.0 ( 5.9) |
| 25 | 2.5 ( 212.9) | 2.2 ( 187.4) | 1.2 ( 102.2) | 0.5 ( 43.4) | 0.9 ( 73.3) | 0.6 ( 51.1) | 0.1 ( 4.4) | 0.1 ( 8.2) | 0.1 ( 11.1) | 0.1 ( 8.5) |
| 20 | 2.4 ( 42.2) | 2.1 ( 36.9) | 1.2 ( 21.1) | 0.1 ( 1.7) | 1.2 ( 21.1) | 0.2 ( 4.0) | 0.0 ( 0.6) | 0.0 ( 0.4) | 0.1 ( 1.5) | 0.0 ( 0.7) |
| 15 | 2.8 ( 4.7) | 1.7 ( 2.9) | 2.3 ( 3.9) | 0.1 ( 0.2) | 2.2 ( 3.7) | 0.2 ( 0.4) | 0.1 ( 0.1) | 0.5 ( 0.9) | 0.2 ( 0.3) | 0.1 ( 0.1) |
| 10 | 3.0 ( 3.7) | 2.3 ( 2.8) | 2.0 ( 2.4) | 0.1 ( 0.1) | 1.4 ( 1.7) | 0.1 ( 0.1) | 0.1 ( 0.1) | 1.2 ( 1.5) | 0.3 ( 0.3) | 0.0 ( 0.0) |

**Table A3.** Error estimate for a polar winter profile during the FR period. Errors are given in pptv (relative errors in %).

| Altitude | total error | noise | total parameter | Gain | LOS | HNO$_4$ | Shift | ILS | Temperature | ClONO$_2$ |
|---|---|---|---|---|---|---|---|---|---|---|
| 40 | 0.0 ( 45.8) | 0.0 ( 34.7) | 0.0 ( 30.5) | 0.0 ( 16.7) | 0.0 ( 20.8) | 0.0 ( 9.3) | 0.0 ( 0.9) | 0.0 ( 7.4) | 0.0 ( 5.8) | 0.0 ( 4.4) |
| 35 | 0.0 ( 46.6) | 0.0 ( 34.6) | 0.0 ( 29.3) | 0.0 ( 16.0) | 0.0 ( 20.0) | 0.0 ( 9.3) | 0.0 ( 0.9) | 0.0 ( 7.3) | 0.0 ( 5.9) | 0.0 ( 4.4) |
| 30 | 0.2 ( 47.8) | 0.2 ( 40.7) | 0.1 ( 26.3) | 0.0 ( 11.7) | 0.1 ( 19.4) | 0.0 ( 10.5) | 0.0 ( 0.7) | 0.0 ( 1.8) | 0.0 ( 4.1) | 0.0 ( 4.1) |
| 25 | 2.4 ( 58.5) | 2.2 ( 53.6) | 1.1 ( 26.8) | 0.4 ( 8.8) | 0.8 ( 19.7) | 0.6 ( 13.6) | 0.0 ( 0.4) | 0.1 ( 2.4) | 0.1 ( 2.9) | 0.2 ( 5.1) |
| 20 | 2.8 ( 22.8) | 2.7 ( 22.0) | 0.9 ( 7.3) | 0.0 ( 0.4) | 0.8 ( 6.8) | 0.3 ( 2.4) | 0.1 ( 0.4) | 0.0 ( 0.1) | 0.0 ( 0.1) | 0.1 ( 1.0) |
| 15 | 4.4 ( 7.7) | 1.8 ( 3.1) | 4.0 ( 7.0) | 0.0 ( 0.1) | 3.9 ( 6.8) | 0.2 ( 0.4) | 0.0 ( 0.0) | 0.9 ( 1.6) | 0.1 ( 0.1) | 0.0 ( 0.1) |
| 10 | 2.7 ( 3.1) | 2.5 ( 2.9) | 0.9 ( 1.0) | 0.2 ( 0.2) | 0.5 ( 0.6) | 0.1 ( 0.1) | 0.1 ( 0.1) | 0.1 ( 0.1) | 0.5 ( 0.6) | 0.1 ( 0.1) |

**Table A4.** Error estimate for an equatorial profile during the RR period. Errors are given in pptv (relative errors in %).

| Altitude | total error | noise | total parameter | Gain | LOS | HNO$_4$ | Shift | ILS | Temperature | ClONO$_2$ |
|---|---|---|---|---|---|---|---|---|---|---|
| 40 | 0.0 ( 3058.9) | 0.0 ( 2867.7) | 0.0 ( 879.4) | 0.0 ( 172.1) | 0.0 ( 124.3) | 0.0 ( 726.5) | 0.0 ( 47.8) | 0.0 ( 372.8) | 0.0 ( 18.2) | 0.0 ( 210.3) |
| 35 | 0.0 (18560.0) | 0.0 (17998.0) | 0.0 ( 5511.9) | 0.0 ( 899.9) | 0.0 ( 899.9) | 0.0 ( 4443.2) | 0.0 ( 303.7) | 0.0 ( 2531.0) | 0.0 ( 146.2) | 0.0 ( 1293.6) |
| 30 | 0.2 ( 73.5) | 0.2 ( 60.7) | 0.1 ( 41.6) | 0.0 ( 13.1) | 0.1 ( 19.5) | 0.0 ( 14.1) | 0.0 ( 2.0) | 0.1 ( 31.3) | 0.0 ( 3.5) | 0.0 ( 3.5) |
| 25 | 2.6 ( 19.9) | 2.0 ( 15.3) | 1.6 ( 12.2) | 0.4 ( 3.2) | 1.2 ( 9.2) | 0.3 ( 2.4) | 0.1 ( 0.5) | 0.9 ( 6.9) | 0.1 ( 0.6) | 0.1 ( 0.5) |
| 20 | 3.3 ( 5.5) | 2.4 ( 4.0) | 2.2 ( 3.7) | 0.6 ( 1.0) | 2.1 ( 3.5) | 0.1 ( 0.1) | 0.1 ( 0.1) | 0.3 ( 0.5) | 0.1 ( 0.2) | 0.0 ( 0.1) |
| 15 | 6.2 ( 7.3) | 5.1 ( 6.0) | 3.6 ( 4.3) | 1.0 ( 1.2) | 3.4 ( 4.0) | 0.4 ( 0.5) | 0.0 ( 0.0) | 0.0 ( 0.0) | 0.5 ( 0.6) | 0.0 ( 0.0) |
| 10 | 6.2 ( 7.3) | 4.9 ( 5.8) | 3.7 ( 4.4) | 1.1 ( 1.3) | 3.5 ( 4.1) | 0.4 ( 0.5) | 0.0 ( 0.0) | 0.1 ( 0.1) | 0.5 ( 0.6) | 0.0 ( 0.1) |

**Table A5.** Error estimate for a polar summer profile during the RR period. Errors are given in pptv (relative errors in %).

| Altitude | total error | noise | total parameter | Gain | LOS | HNO$_4$ | Shift | ILS | Temperature | ClONO$_2$ |
|---|---|---|---|---|---|---|---|---|---|---|
| 40 | 0.0 ( 336.8) | 0.0 ( 307.1) | 0.0 ( 158.5) | 0.0 ( 96.1) | 0.0 ( 56.5) | 0.0 ( 73.3) | 0.0 ( 2.2) | 0.0 ( 70.3) | 0.0 ( 2.7) | 0.0 ( 12.9) |
| 35 | 0.0 ( 333.4) | 0.0 ( 296.4) | 0.0 ( 148.2) | 0.0 ( 92.6) | 0.0 ( 55.6) | 0.0 ( 72.2) | 0.0 ( 2.0) | 0.0 ( 67.6) | 0.0 ( 2.7) | 0.0 ( 13.0) |
| 30 | 0.2 ( 299.3) | 0.2 ( 273.3) | 0.1 ( 123.6) | 0.1 ( 80.7) | 0.0 ( 52.1) | 0.1 ( 69.0) | 0.0 ( 0.4) | 0.0 ( 27.3) | 0.0 ( 3.1) | 0.0 ( 7.5) |
| 25 | 2.2 ( 72.1) | 2.1 ( 68.9) | 0.6 ( 19.3) | 0.3 ( 10.2) | 0.1 ( 2.9) | 0.5 ( 15.7) | 0.0 ( 0.6) | 0.0 ( 1.0) | 0.0 ( 0.5) | 0.1 ( 1.9) |
| 20 | 3.0 ( 16.2) | 2.2 ( 11.9) | 2.0 ( 10.8) | 0.0 ( 0.1) | 2.0 ( 10.8) | 0.1 ( 0.4) | 0.1 ( 0.5) | 0.4 ( 2.3) | 0.0 ( 0.2) | 0.0 ( 0.1) |
| 15 | 2.8 ( 3.9) | 2.2 ( 3.1) | 1.8 ( 2.5) | 0.2 ( 0.3) | 1.6 ( 2.3) | 0.1 ( 0.2) | 0.0 ( 0.0) | 0.8 ( 1.2) | 0.0 ( 0.1) | 0.0 ( 0.0) |
| 10 | 3.0 ( 3.6) | 1.8 ( 2.2) | 2.5 ( 3.0) | 0.2 ( 0.3) | 2.2 ( 2.6) | 0.0 ( 0.1) | 0.1 ( 0.2) | 1.0 ( 1.2) | 0.1 ( 0.1) | 0.0 ( 0.0) |

**Table A6.** Error estimate for a polar winter profile during the RR period. Errors are given in pptv (relative errors in %).

| Altitude | total error | noise | total parameter | Gain | LOS | HNO$_4$ | Shift | ILS | Temperature | ClONO$_2$ |
|---|---|---|---|---|---|---|---|---|---|---|
| 40 | 0.0 ( 632.5) | 0.0 ( 367.3) | 0.0 ( 510.1) | 0.0 ( 204.0) | 0.0 ( 448.9) | 0.0 ( 67.3) | 0.0 ( 9.8) | 0.0 ( 24.5) | 0.0 ( 61.2) | 0.0 ( 36.7) |
| 35 | 0.0 ( 608.6) | 0.0 ( 342.4) | 0.0 ( 494.5) | 0.0 ( 190.2) | 0.0 ( 437.4) | 0.0 ( 66.6) | 0.0 ( 9.5) | 0.0 ( 22.8) | 0.0 ( 60.9) | 0.0 ( 36.1) |
| 30 | 0.2 ( 369.8) | 0.1 ( 228.9) | 0.2 ( 281.8) | 0.1 ( 112.7) | 0.1 ( 264.1) | 0.0 ( 42.3) | 0.0 ( 6.0) | 0.0 ( 2.5) | 0.0 ( 33.5) | 0.0 ( 22.9) |
| 25 | 2.9 ( 308.3) | 2.2 ( 233.9) | 1.8 ( 191.3) | 0.7 ( 76.5) | 1.6 ( 170.1) | 0.4 ( 41.5) | 0.1 ( 6.1) | 0.2 ( 26.6) | 0.2 ( 20.2) | 0.2 ( 23.4) |
| 20 | 2.9 ( 46.0) | 2.7 ( 42.8) | 1.1 ( 17.4) | 0.1 ( 1.4) | 1.0 ( 15.9) | 0.2 ( 2.5) | 0.1 ( 1.2) | 0.3 ( 4.6) | 0.1 ( 0.9) | 0.1 ( 1.3) |
| 15 | 3.4 ( 5.1) | 2.3 ( 3.4) | 2.5 ( 3.7) | 0.3 ( 0.5) | 2.4 ( 3.6) | 0.1 ( 0.2) | 0.0 ( 0.1) | 0.5 ( 0.7) | 0.1 ( 0.1) | 0.0 ( 0.0) |
| 10 | 2.2 ( 2.6) | 1.5 ( 1.8) | 1.6 ( 1.9) | 0.0 ( 0.0) | 1.4 ( 1.7) | 0.1 ( 0.1) | 0.0 ( 0.0) | 0.7 ( 0.9) | 0.2 ( 0.2) | 0.0 ( 0.0) |

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
