# Peer review of "MIPAS IMK/IAA Carbon Tetrachloride (CCl4) Retrieval and first Comparison with other Instruments"

_Atmospheric Measurement Techniques, 2016_

## Referee Comment (RC1) · H. C. Pumphrey (Referee) · 28 Feb 2017

**General remarks**

This paper describes a dataset consisting of CCl$_4$ mixing ratios retrieved from radiances from the MIPAS instrument on ENVISAT. The data are a valuable contribution to our understanding of stratospheric chemistry and the paper is well written and organised. It should therefore be published with only technical corrections. I list some of these corrections below; they mostly concern the figures, many of which are too small in the form presented for review and are such that the journal's page layout staff will not be able to make them much larger in the final version.

[Figure]

**Specific corrections**

- Page 3, Line 87: Figures should be referenced in numerical order: the first reference to a figure in the text should be to figure 1. (I would not re-order the figures to correct this, rather, I would remove the reference to figure 3 in this line.)

- Figure 1 and Figure 3: The figures are really too small. They might be just about OK if made the full width of the two columns in the final version, but I am not sure about this. Text on figures should be a similar size to the text in the figure caption. The legends in these figures are particularly hard to read; it would help if they were not on top of the curves.

- Figure 2: It is good that the two panels have the same colour scale for easy comparison. But the scale should go down to the lowest values shown in the left panel. As the figure stands, the colour white is used to represent two distinct things: negative values, and areas where there are no data. The "no data" areas should be left white, and the (white free) colour scale should extend so that it applies to the negative areas. The figure title "PAN, 200807" is not needed to distinguish the two panels and tells the reader nothing that is not in the caption. It should be removed. The contour lines should be in a colour (or colours) that allow them to be seen against the colour scale. With these changes made, Page 3 Line 113 will need to be changed (and can possibly be simplified). Many of these comments also apply to figures 4, 5 and 6.

- Page 6 line 118: "setup" should be "set up" because it is a verb. (Note that on page 7 line 140, "setup" is a noun and should be left as it is.)

- Page 6 line 131: "The spectral region [. . . ] could be narrowed to [values]" Narrowed from what?

- Figure 7: This figure is probably OK if shown at the full 8.3cm width of a journal column. The title "Rows of A" on each panel should be removed.

- Page 11 sec 5.1.1: It would be nice to add a figure showing the mean ATMOS profiles and a suitably-averaged MIPAS profile for the same time of year.

- Page 11 line 212: Remove comma after "profile".

- Page 13 figure 9: It is again marginal whether this figure is large enough. It is probably OK at the full two-column width of the journal page. It would be a great improvement if the five panels were labelled (a) to (e) so that the text could refer to "panel (e)" rather than "second panel to the right". (I think this means "second panel *from* the right" but I am not sure; letter labels would remove this kind of confusion.)

- Figures 10 and 11: I again have concerns about the sizing of the figures. The text in the captions is unreadably small at the size of the review article. The title on Figure 11 is not needed.

---

## Referee Comment (RC2) · Anonymous Referee #2 · 2 Mar 2017

This manuscript provides details about a new approach specifically developed to retrieve vertical profile distributions of carbon tetrachloride on the global scale from MIPAS level-1b limb emission spectra, recorded on-board ENVISAT over the 2002-2012 time period, first at full resolution (FR), then at reduced resolution (RR).

As for previous studies, a broad micro-window centered on ~789 cm-1 is selected, but the authors underscore the influence of PAN on the retrieval, note that while they account for line-mixing effects on the strong CO2 Q-branch at 791.5 cm-1, the fitting of this feature is not entirely satisfactory and that it is preferable to omit the corresponding "spectral slice".

This approach is used to retrieve CCl4 from a subset of the available MIPAS observations, and profiles are retrieved for 09/2003 (FR), 07/2008, 01/2010, 03 and 04/2011 (RR), but it is not clear why these periods have been selected.

[Figure]

The satellite product is carefully characterized in terms of information content and a complete uncertainty analysis is provided, indicating that reliable measurements are available in the 8-25 km altitude range on the global scale, with a vertical resolution which is dependent on (and decreasing with) altitude.

Finally, the new CCl4 cross-section parameters of Harrison et al. (JQSRT, 186, 2017) are tested, and some intercomparisons with collocated (or historical) measurements are presented.

Overall, this is a good manuscript, well-structured and clear, although some figures are desperately small in the present version. It is a useful contribution for a species subject of attention (e.g., SPARC, 2016). The paper is appropriate for AMT and I would recommend publication after implementation of a few changes.

Major comments

- The last sentence of the abstract should be removed, it states "The decline in CCl4 abundance during the MIPAS Envisat measurement period (July 2002 to April 2012) is clearly reflected in the retrieved distributions". I agree that information on (and a proper quotation of) the CCl4 trend would have been a very valuable addition to this study, BUT only a subset of the observations is presented, the periods shown do not cover the 10-year time interval (09/2003 – 04/2011 instead of 07/2002 – 04/2012) and the reader has no element to gauge the CCl4 rate of change and to judge about the validity of this assertion

- Figure 2 shows that the PAN product jointly retrieved with CCl4 is superior to the standard PAN data available thus far from the MIPAS team, it would be equally important to have an idea of the impact of retrieving versus neglecting PAN on the quality of the CCl4 product! In particular, is there a systematic impact on the CCl4 mixing ratios, allowing to close the well-known gap between in situ and remote-sensing data (see e.g. Chipperfield et al., ACP, 16, 2016)? This information would be very valuable for the community and I suggest adding two panels to Fig.2 dedicated to CCl4 with/without

PAN

Specific comments

- Section 4.2: it is somewhat strange that the FR measurements provide a lower DOF (3.5) than the RR observations (4.0). What could be the reason for this? This deserves a comment.

- Figure 7 is really small and the y-axis unnecessarily goes up to 80 km, I suggest limiting the altitude range to something like 0-50 km to improve readability

- Section 5.1.1: ATMOS results are used for a qualitative comparison, but still, why did you use profiles retrieved in the mid-1980s by Zander et al, when the CCl4 spectroscopy was of poor quality? (see Brown et al., Appl. Opt., 35, 1996). Results reported later on by Zander et al. (e.g. GRL, 23, 1996) are very likely more appropriate for a sensible comparison. An alternative would be to use the ATMOS version 3 results available from http://remus.jpl.nasa.gov/atmos/atmosversion3/atmosversion3.html and fully described in Irion et al. (Appl. Opt., 41, 2002)

- Section 5.2.1: the agreement between ACE and MIPAS is best below 15 km (lines 265-266 on page 13), but this is also mostly where the number of coincidences is the smallest (second left frame of Fig. 9). Could this inconsistent sampling have an impact on the statistics?

Minor comments and typos

- The title is not very informative; it could be edited to inform about the fact that first intercomparisons are included in this work

-Page 2, line 22: "in 1987, when it was restricted": this is incorrect, CCl4 was not among the first species controlled under the Montreal Protocol, it was added to the list in the 1990 London Amendment

- Page 2, line 28: these top-down emissions were evaluated instead of "reported"

- Page 2, line 29, I think a comma is needed after "unreported"

- Page 2, lien 35, here, I suggest replacing "considerably" by "now"

- Page 2, line 37: I would remove the reference to MIPAS here ("besides those of MIPAS...", it is appropriate to introduce the new measurements later on, after the review of previous works

- Page 3, line 65, "as reduced" instead of "is reduced"

- Page 3, line 85: the information about the actual spectral range fitted to retrieve CCl4 is not consistent across the manuscript (see table 1, end of section 3.2...), this should be fixed

- Caption of Fig.2: I guess that the "Black: measured spectrum, hardly discernible because overplotted by modelled spectra" warning has nothing to do here...

- Page 7, line 152: I would edit to "of CCl4 for different time periods. All of the..."

- Section 5.1.1.: ATMOS also participated to three other shuttle missions, in 1992, 1993 and 1994.

- Section 5.2.: please reword to something like "Since all collocated measurements were retrieved using the spectroscopic data of Nemtchinov and Varanasi (2003) introduced in HITRAN 2000, MIPAS Envisat retrievals based on the same spectroscopic dataset were also used for consistency and in order not to mask possible other discrepancies."

---

## Author Comment (AC1) · 27 Apr 2017

The authors thank the Reviewers for their revision of our manuscript and their helpful comments. In the following the original comments are inserted in *italic face* while our replies are printed in normal face.

**Reviewer 1:**
*Page 3, Line 87: Figures should be referenced in numerical order: the first reference to a figure in the text should be to figure 1. (I would not re-order the figures to correct this, rather, I would remove the reference to figure 3 in this line.)*
**Reply 1:** Agreed. This will be changed.

*Figure 1 and Figure 3: The figures are really too small. They might be just about OK if made the full width of the two columns in the final version, but I am not sure about*

*this. Text on figures should be a similar size to the text in the figure caption. The
legends in these figures are particularly hard to read; it would help if they were not on
top of the curves.*
**Reply 2:** Agreed. Figure 1 and figure 3 will be split up into two individual figures which
each span the whole width of the page. The text will be changed accordingly.

*Figure 2: It is good that the two panels have the same colour scale for easy
comparison. But the scale should go down to the lowest values shown in the left
panel. As the figure stands, the colour white is used to represent two distinct things:
negative values, and areas where there are no data. The "no data" areas should be
left white, and the (white free) colour scale should extend so that it applies to the
negative areas. The figure title "PAN, 200807" is not needed to distinguish the two
panels and tells the reader nothing that is not in the caption. It should be removed.
The contour lines should be in a colour (or colours) that allow them to be seen against
the colour scale. With these changes made, Page 3 Line 113 will need to be changed
(and can possibly be simplified). Many of these comments also apply to figures 4, 5
and 6.*
**Reply 3:** Agreed. The colour scale will be changed to cover the whole range of values
of both figures. The titles will be removed. A decision on whether or not the contour
lines will be changed to white or left black will be made depending on how dark the
background colours will be. The text will be changed accordingly to reflect the
changed colour scale.

*Page 6 line 118: "setup" should be "set up" because it is a verb. (Note that on page 7
line 140, "setup" is a noun and should be left as it is.)*
**Reply 4:** Agreed. This will be corrected.

*Page 6 line 131: "The spectral region [...] could be narrowed to [values]" Narrowed
from what?*

**Reply 5:** The gap could be narrowed down to 791.0 to 792.0 cm$^{-1}$ from 790.5 to 792.5 cm$^{-1}$. Additional information on this will be given in the revised version of the manuscript.

*Figure 7: This figure is probably OK if shown at the full 8.3cm width of a journal column. The title "Rows of A" on each panel should be removed.*
**Reply 6:** We will change the figures to better reflect the altitude region of interest. The titles will be removed and the figures are intended to span the entire width of the page.

*Page 11 sec 5.1.1: It would be nice to add a figure showing the mean ATMOS profiles and a suitably-averaged MIPAS profile for the same time of year.*
**Reply 7:** We intend to provide the suggested figure in the revised version of the manuscript. Average MIPAS profiles of the respective region and time of year will be used. The text will be changed accordingly.

*Page 11 line 212: Remove comma after "profile".*
**Reply 8:** Agreed. The comma will be removed.

*Page 13 figure 9: It is again marginal whether this figure is large enough. It is probably OK at the full two-column width of the journal page. It would be a great improvement if the five panels were labelled (a) to (e) so that the text could refer to "panel (e)" rather than "second panel to the right". (I think this means "second panel from the right" but I am not sure; letter labels would remove this kind of confusion.)*
**Reply 9:** Agreed. This will be changed. The text will be changed accordingly.

*Figures 10 and 11: I again have concerns about the sizing of the figures. The text in the captions is unreadably small at the size of the review article. The title on Figure 11 is not needed.*
**Reply 10:** Agreed. This will be changed. The figures will each span the full width of
the document. Fig. 10 will be split up into two individual figures. The text will be changed accordingly.

**Reviewer 2:**
**Major comments**
*The last sentence of the abstract should be removed, it states ?The decline in CCl4 abundance during the MIPAS Envisat measurement period (July 2002 to April 2012) is clearly reflected in the retrieved distributions?. I agree that information on (and a proper quotation of) the CCl4 trend would have been a very valuable addition to this study, BUT only a subset of the observations is presented, the periods shown do not cover the 10-year time interval (09/2003 ? 04/2011 instead of 07/2002 ? 04/2012) and the reader has no element to gauge the CCl4 rate of change and to judge about the validity of this assertion*
**Reply 1:** Trends have now been estimated from the full data set and was included in the paper. A subsection will be added to discuss the results of the trend estimation. The according text is going to state good agreement with the trends estimated by Valeri et al. (2017).

*Figure 2 shows that the PAN product jointly retrieved with CCl4 is superior to the standard PAN data available thus far from the MIPAS team, it would be equally important to have an idea of the impact of retrieving versus neglecting PAN on the quality of the CCl4 product! In particular, is there a systematic impact on the CCl4 mixing ratios, allowing to close the well-known gap between in situ and remote-sensing data (see e.g. Chipperfield et al., ACP, 16, 2016)? This information would be very valuable for the community and I suggest adding two panels to Fig.2 dedicated to CCl4 with/without*
**Reply 2:** We believe there is a misunderstanding here. None of the two figures show the PAN results for $CCl_4$ being left out entirely in the retrieval. $CCl_4$ was accounted for in the MIPAS retrieval before the gas was an actual target of the retrieval itself.

However, optimizing the retrieval for $CCl_4$ led to changes in the PAN distributions. The influence of these changes are reflected in the two panels of the figure. Since both species, PAN and $CCl_4$, were accounted for in the original PAN retrieval, we do not see a benefit from showing $CCl_4$ results without PAN. Fig. 2 is supposed to ensure that changes made to the retrieval to optimize it for $CCl_4$ did not decrease the quality of the PAN results. Fig. 2 proves that, on the contrary, these changes also led to improvement of the PAN results.

*Section 4.2: it is somewhat strange that the FR measurements provide a lower DOF (3.5) than the RR observations (4.0). What could be the reason for this? This deserves a comment.*
**Reply 3:** We don't think that this is strange, because the RR measurements have a finer altitude sampling. Measurements were taken at 27 instead of 17 tangent altitudes during the RR and FR period, respectively. This easily explains the higher DOF of the RR observations. A sentence will be added for clarification.

*Figure 7 is really small and the y-axis unnecessarily goes up to 80 km, I suggest limiting the altitude range to something like 0-50 km to improve readability*
**Reply 4:** Agreed. The altitude range will be limited to 0-40 km and the figure will span the whole page of the journal to improve legibility.

*Section 5.1.1: ATMOS results are used for a qualitative comparison, but still, why did you use profiles retrieved in the mid-1980s by Zander et al, when the CCl4 spectroscopy was of poor quality? (see Brown et al., Appl. Opt., 35, 1996). Results reported later on by Zander et al. (e.g. GRL, 23, 1996) are very likely more appropriate for a sensible comparison. An alternative would be to use the ATMOS version 3 results available from http://remus.jpl.nasa.gov/atmos/atmosversion3/atmosversion3.html and fully described in Irion et al. (Appl. Opt., 41, 2002)*

**Reply 5:** Agreed. The qualitative comparison will be performed with the results reported by Zander et al. (1996).

*Section 5.2.1: the agreement between ACE and MIPAS is best below 15 km (lines 265-266 on page 13), but this is also mostly where the number of coincidences is the smallest (second left frame of Fig. 9). Could this inconsistent sampling have an impact on the statistics?*
**Reply 6:** Since the comparison is based on coincident measurements, the impact of inconsistent sampling should be negligible.

**Minor comments and typos**
*The title is not very informative; it could be edited to inform about the fact that first intercomparisons are included in this work*
**Reply 7:** Agreed. The title will be changed to "MIPAS IMK/IAA Carbon Tetrachloride (CCl$_4$) Retrieval and first Comparison with other Instruments".

*Page 2, line 22: "in 1987, when it was restricted": this is incorrect, CCl4 was not among the first species controlled under the Montreal Protocol, it was added to the list in the 1990 London Amendment*
**Reply 8:** Agreed. This will be changed.

*Page 2, line 28: these top-down emissions were evaluated instead of "reported"*
**Reply 9:** Agreed. This will be changed.

*Page 2, line 29, I think a comma is needed after "unreported"*
**Reply 10:** In case the reviewer is referring to "Even when possible CCl$_4$ precursors and unreported inadvertent emissions are accounted for..." we don't think a comma is needed after "unreported".

*Page 2, lien 35, here, I suggest replacing "considerably" by "now"*
**Reply 11:** Agreed. This will be changed.

*Page 2, line 37: I would remove the reference to MIPAS here ("besides those of MIPAS...", it is appropriate to introduce the new measurements later on, after the review of previous works*
**Reply 12:** Agreed. This will be changed.

*Page 3, line 65, "as reduced" instead of "is reduced"*
**Reply 13:** Agreed. This will be corrected.

*Page 3, line 85: the information about the actual spectral range fitted to retrieve CCl4 is not consistent across the manuscript (see table 1, end of section 3.2...), this should be fixed*
**Reply 14:** Agreed. This will be corrected.

*Caption of Fig.2: I guess that the "Black: measured spectrum, hardly discernible because overplotted by modelled spectra" warning has nothing to do here...*
**Reply 15:** Agreed. This sentence will be moved to Fig. 3.

*Page 7, line 152: I would edit to "of CCl4 for different time periods. All of the.."*
**Reply 16:** Agreed. This will be added.

*Section 5.1.1.: ATMOS also participated to three other shuttle missions, in 1992, 1993 and 1994.*
**Reply 17:** Agreed. This will be included in the text.

*Section 5.2.: please reword to something like "Since all collocated measurements were retrieved using the spectroscopic data of Nemtchinov and Varanasi (2003)*

*introduced in HITRAN 2000, MIPAS Envisat retrievals based on the same
spectroscopic dataset were also used for consistency and in order not to mask
possible other discrepancies."*
**Reply 18:** Agreed. This will be changed.

**References**

Valeri, M., Barbara, F., Boone, C., Ceccherini, S., Gai, M., Maucher, G., Raspollini, P., Ridolfi, M., Sgheri, L., Wetzel, G., and Zoppetti, N. (2017). Ccl$_4$ distribution derived from mipas esa v7 data: validation, trend and lifetime estimation. *Atmospheric Chemistry and Physics Discussions*, 2017:1–31.

Zander, R., Mahieu, E., Gunson, M. R., Abrams, M. C., Chang, A. Y., Abbas, M. M., Aelig, C., Engel, A., Goldman, A., Irion, F. W., Kämpfer, N., Michelson, H. A., Newchurch, M. J., Rinsland, C. P., Salawitch, R. J., Stiller, G. P., and Toon, G. C. (1996). The 1994 northern midlatitude budget of stratospheric chlorine derived from ATMOS/ATLAS–3 observations. *Geophys. Res. Lett.*, 23(17):2357–2360.

---

## Author Response (AR1)

The authors thank the Reviewers for their revision of our manuscript and their helpful comments. In the following the original comments are inserted in *italic face* while our replies are printed in normal face.

**Reviewer 1:**
*Page 3, Line 87: Figures should be referenced in numerical order: the first reference to a figure in the text should be to figure 1. (I would not re-order the figures to correct this, rather, I would remove the reference to figure 3 in this line.)*
**Reply 1:** Agreed and changed.

*Figure 1 and Figure 3: The figures are really too small. They might be just about OK if made the full width of the two columns in the final version, but I am not sure about this. Text on figures should be a similar size to the text in the figure caption. The legends in these figures are particularly hard to read; it would help if they were not on top of the curves.*
**Reply 2:** Agreed. Figure 1 and figure 3 were split up in two individual figures which now span the full width of the page. The text was changed accordingly.

*Figure 2: It is good that the two panels have the same colour scale for easy comparison. But the scale should go down to the lowest values shown in the left panel. As the figure stands, the colour white is used to represent two distinct things: negative values, and areas where there are no data. The "no data" areas should be left white, and the (white free) colour scale should extend so that it applies to the negative areas. The figure title "PAN, 200807" is not needed to distinguish the two panels and tells the reader nothing that is not in the caption. It should be removed. The contour lines should be in a colour (or colours) that allow them to be seen against the colour scale. With these changes made, Page 3 Line 113 will need to be changed (and can possibly be simplified). Many of these comments also apply to figures 4, 5 and 6.*
**Reply 3:** Agreed. The colour scale was changed to cover the whole range of values of both figures. The titles have been removed. We did not change the contour line colour, because the background colours are now much lighter and white contour lines would be harder to see than black ones. The text was changed accordingly to reflect the updated colour scale.

*Page 6 line 118: "setup" should be "set up" because it is a verb. (Note that on page 7 line 140, "setup" is a noun and should be left as it is.)*
**Reply 4:** Agreed and corrected.

*Page 6 line 131: "The spectral region [...] could be narrowed to [values]" Narrowed from what?*
**Reply 5:** The gap could be narrowed down to 791.0 to $792.0\,\mathrm{cm}^{-1}$ from 790.5 to $792.5\,\mathrm{cm}^{-1}$. Additional information on this is now given in the manuscript.

*Figure 7: This figure is probably OK if shown at the full 8.3cm width of a journal column. The title "Rows of A" on each panel should be removed.*
**Reply 6:** We changed the figures to better reflect the altitude region of interest. The titles were removed and the figures now span the entire width of the page.

*Page 11 sec 5.1.1: It would be nice to add a figure showing the mean ATMOS profiles and a suitably-averaged MIPAS profile for the same time of year.*

**Reply 7:** We now provide the suggested figure in the revised version of the manuscript. Average MIPAS profiles of the respective region and time of year are used. The text was changed accordingly.

*Page 11 line 212: Remove comma after "profile".*
**Reply 8:** Agreed and removed.

*Page 13 figure 9: It is again marginal whether this figure is large enough. It is probably OK at the full two-column width of the journal page. It would be a great improvement if the five panels were labelled (a) to (e) so that the text could refer to "panel (e)" rather than "second panel to the right". (I think this means "second panel from the right" but I am not sure; letter labels would remove this kind of confusion.)*
**Reply 9:** Agreed and changed. The text was changed accordingly.

*Figures 10 and 11: I again have concerns about the sizing of the figures. The text in the captions is unreadably small at the size of the review article. The title on Figure 11 is not needed.*
**Reply 10:** Agreed and changed. Fig. 10 was split up into two individual figures. These figures now each span the full width of the document. The text was changed accordingly.

**Reviewer 2:**
**Major comments**
*The last sentence of the abstract should be removed, it states "The decline in CCl4 abundance during the MIPAS Envisat measurement period (July 2002 to April 2012) is clearly reflected in the retrieved distributions". I agree that information on (and a proper quotation of) the CCl4 trend would have been a very valuable addition to this study, BUT only a subset of the observations is presented, the periods shown do not cover the 10-year time interval (09/2003 - 04/2011 instead of 07/2002 - 04/2012) and the reader has no element to gauge the CCl4 rate of change and to judge about the validity of this assertion*
**Reply 1:** Trends have now been estimated from the full data set and an respective figure was included in the paper. A subsection was added to discuss the results of the trend estimation. The according text states good agreement with the trends estimated by Valeri et al. (2017).

*Figure 2 shows that the PAN product jointly retrieved with CCl4 is superior to the standard PAN data available thus far from the MIPAS team, it would be equally important to have an idea of the impact of retrieving versus neglecting PAN on the quality of the CCl4 product! In particular, is there a systematic impact on the CCl4 mixing ratios, allowing to close the well-known gap between in situ and remote-sensing data (see e.g. Chipperfield et al., ACP, 16, 2016)? This information would be very valuable for the community and I suggest adding two panels to Fig.2 dedicated to CCl4 with/without*
**Reply 2:** We believe there is a misunderstanding here. None of the two figures show the PAN results for $CCl_4$ being left out entirely in the retrieval. $CCl_4$ was accounted for in the MIPAS retrieval before the gas was an actual target of the retrieval itself. However, optimizing the retrieval for $CCl_4$ led to changes in the PAN distributions. The influence of these changes are reflected in the two panels of Fig. 2. Since both species, PAN and $CCl_4$, were accounted for in the original PAN retrieval, we do not see a benefit from showing $CCl_4$ results without PAN. Fig. 2 is supposed to ensure that changes made to the retrieval to optimize it for $CCl_4$ did not decrease the

quality of the PAN results. Fig. 2 proves that, on the contrary, these changes also led to improvement of the PAN results.

*Section 4.2: it is somewhat strange that the FR measurements provide a lower DOF (3.5) than the RR observations (4.0). What could be the reason for this? This deserves a comment.*
**Reply 3:** We don't think this is strange, because the RR measurements have a finer altitude sampling. Measurements were taken at 27 instead of 17 tangent altitudes during the RR and FR period, respectively. This easily explains the higher DOF of the RR observations. A sentence was added for clarification.

*Figure 7 is really small and the y-axis unnecessarily goes up to 80 km, I suggest limiting the altitude range to something like 0-50 km to improve readability*
**Reply 4:** Agreed. The altitude range was limited to 0-40 km and the figure is now spanning the full width of the page to improve legibility.

*Section 5.1.1: ATMOS results are used for a qualitative comparison, but still, why did you use profiles retrieved in the mid-1980s by Zander et al, when the CCl4 spectroscopy was of poor quality? (see Brown et al., Appl. Opt., 35, 1996). Results reported later on by Zander et al. (e.g. GRL, 23, 1996) are very likely more appropriate for a sensible comparison. An alternative would be to use the ATMOS version 3 results available from http://remus.jpl.nasa.gov/atmos/atmosversion3/atmosversion3.html and fully described in Irion et al. (Appl. Opt., 41, 2002)*
**Reply 5:** Agreed. A figure regarding the qualitative comparison was added to the manuscript. We are now using results reported by Zander et al. (1996).

*Section 5.2.1: the agreement between ACE and MIPAS is best below 15 km (lines 265-266 on page 13), but this is also mostly where the number of coincidences is the smallest (second left frame of Fig. 9). Could this inconsistent sampling have an impact on the statistics?*
**Reply 6:** Since the comparison is based on coincident measurements, the impact of inconsistent sampling should be negligible.

**Minor comments and typos**
*The title is not very informative; it could be edited to inform about the fact that first intercomparisons are included in this work*
**Reply 7:** Agreed. The title was changed to "MIPAS IMK/IAA Carbon Tetrachloride ($CCl_4$) Retrieval and first Comparison with other Instruments".

*Page 2, line 22: "in 1987, when it was restricted": this is incorrect, CCl4 was not among the first species controlled under the Montreal Protocol, it was added to the list in the 1990 London Amendment*
**Reply 8:** Agreed and changed.

*Page 2, line 28: these top-down emissions were evaluated instead of "reported"*
**Reply 9:** Agreed and changed.

*Page 2, line 29, I think a comma is needed after "unreported"*

**Reply 10:** Agreed and changed.

*Page 2, lien 35, here, I suggest replacing "considerably" by "now"*
**Reply 11:** Agreed and changed.

*Page 2, line 37: I would remove the reference to MIPAS here ("besides those of MIPAS...", it is appropriate to introduce the new measurements later on, after the review of previous works*
**Reply 12:** Agreed and changed.

*Page 3, line 65, "as reduced" instead of "is reduced"*
**Reply 13:** Agreed and corrected.

*Page 3, line 85: the information about the actual spectral range fitted to retrieve CCl4 is not consistent across the manuscript (see table 1, end of section 3.2...), this should be fixed*
**Reply 14:** Agreed and corrected.

*Caption of Fig.2: I guess that the "Black: measured spectrum, hardly discernible because overplotted by modelled spectra" warning has nothing to do here...*
**Reply 15:** Agreed and removed.

*Page 7, line 152: I would edit to "of CCl4 for different time periods. All of the.."*
**Reply 16:** Agreed and added.

*Section 5.1.1.: ATMOS also participated to three other shuttle missions, in 1992, 1993 and 1994.*
**Reply 17:** Agreed. This information is now included in the text.

*Section 5.2.: please reword to something like "Since all collocated measurements were retrieved using the spectroscopic data of Nemtchinov and Varanasi (2003) introduced in HITRAN 2000, MIPAS Envisat retrievals based on the same spectroscopic dataset were also used for consistency and in order not to mask possible other discrepancies."*
**Reply 18:** Agreed and changed.

*Correspondence to:* E. Eckert (ellen.eckert@kit.edu)

**Abstract.** MIPAS thermal limb emission measurements were
used to derive vertically resolved profiles of carbon tetra-
chloride (CCl$_4$). Level-1b data versions MIPAS/5.02 to MI-
PAS/5.06 were converted into volume mixing ratio profiles
using the level-2 processor developed at Karlsruhe Insti-
tute of Technology (KIT) Institute of Meteorology and Cli-
mate Research (IMK) and Consejo Superior de Investiga-

[revised manuscript text omitted]
 ~~(white areas in extratropical regions above ~15 km in thein the original retrievalsPAN results from the joint fitstheclimatological the oldprofiles~~distributions.

**3.2 Line mixing**

Since the spectral region where CCl$_4$ is retrievable contains a CO$_2$ Q-branch, the retrieval is  set up to account for line mixing (Funke et al., 1998). This was done by using the Rosenkranz approximation (Rosenkranz, 1975). Tests were also performed using the computationally more demanding direct diagonalisation, but this approach was not found to noticeably change the results of the retrieval. This is possibly the case because the microwindows were carefully selected to omit major spectral signatures of the CO$_2$ Q-branch and because the effect of line mixing is generally smaller at stratospheric pressure levels. However, it was still necessary to omit parts of the CO$_2$ Q-branch. Fig.  4 and Fig. 5 show spectra where the full spectral region was fitted. In Fig. 4, line mixing was not considered and thus a large peak in the residual is visible close to 791.0 cm$^{-1}$. In Fig. 5, the Rosenkranz approximation was used to account for line mixing. Even though the residual is considerably smaller than without line mixing taken into account - as would be expected - peaks significantly larger than for the remainder of the window are

still visible between 791.0 and 792.0 cm$^{-1}$. Although inclusion of line mixing significantly reduces the residuals in the CO$_2$ branch, the residuals are still unacceptably large there. With the Rosenkranz approximation, however, the spectral region excluded from the fit could be narrowed to 791.0 to 792.0 cm$^{-1}$ from 790.5 to 792.5 cm$^{-1}$.

**3.3 New CCl$_4$ Spectroscopic Data**

During the ongoing development of the MIPAS Envisat CCl$_4$ retrieval, a new CCl$_4$ spectroscopic dataset was published by Harrison et al. (2017). Fig. 6 shows the influence of these spectroscopic data on an altitude-latitude cross-section of CCl$_4$ distributions of July 2008. The upper panel shows what  stratospheric CCl$_4$  distributions retrieved with the original spectroscopic dataset as presented in HITRAN 2000 (Nemtchinov and Varanasi, 2003)  look like. The lower panel shows the same cross-section, but using the new spectroscopic dataset by Harrison et al. (2017) for an otherwise identical retrieval setup. While the qualitative and morphological features of the distribution are very similar, lower volume mixing ratios of CCl$_4$ result when the new spectroscopic  dataset is used. Comparing these with reported values of ground based measurements as presented in SPARC (2016) indicates that the updated spectroscopic data  lead to results which, in the tropopause region, agree better with tropospheric measurements. Tropospheric volume mixing ratios are reported to be at approximately 95 pptv which is very close to what MIPAS Envisat presents around the tropical tropopause and at mid-latitudes of the northern hemisphere when using the new spectroscopic dataset. In contrast, using HITRAN 2000 sometimes results in volume mixing ratios above 100 pptv in the same region. Thus, we consider the new spectroscopic dataset more adequate for the retrieval of CCl$_4$.

**4 Results**

**4.1 Distributions**

Fig. 7, the lower panel of Fig. 6 and Fig. 8 give an overview of the latitudinal and altitude distribution of CCl$_4$ of different time periods. All of the altitude-latitude cross-sections show the expected pattern of CCl$_4$ with a rapid decrease with increasing altitude in the stratosphere, as the gas is photolyzed there. In addition, highest volume mixing ratios appear at the equator where CCl$_4$, along with many other trace gases, enters the stratosphere due to the upward transport associated with the Brewer-Dobson circulation. During January 2010, March 2011 and particularly April 2011, subsidence of higher stratospheric air results in reduced mixing ratios over the North pole. In Spring 2011, an unusually stable northern polar vortex resulted in severe ozone depletion and particularly strong subsidence (Manney et al., 2011; Sinnhuber et al., 2011) which is reflected by the observations

[Figure]

**Figure 4.** Impact of the CO$_2$ Q-branch at 11.5 km altitude without considering line mixing(left) and with taking it into account (right). Top panelspanel: spectra; bottom panelspanel: residuals. Note the different scale of the residual axisBlack: 
[revised manuscript text omitted]

Zander, R., Rinsland, C. P., Farmer, C. B., and Norton, R. H.: Infrared spectroscopic measurements of halogenated source gases in the stratosphere with the ATMOS instrument, J. Geophys. Res., 92, 9836–9850, 1987.

Zander, R., Mahieu, E., Gunson, M. R., Abrams, M. C., Chang, A. Y., Abbas, M. M., Aelig, C., Engel, A., Goldman, A., Irion, F. W., Kämpfer, N., Michelson, H. A., Newchurch, M. J., Rinsland, C. P., Salawitch, R. J., Stiller, G. P., and Toon, G. C.: The 1994 northern midlatitude budget of stratospheric chlorine derived from ATMOS/ATLAS–3 observations, Geophys. Res. Lett., 23, 2357–2360, 1996.

---

## Author Response (AR2)

The authors thank the Reviewers for their revision of our manuscript and their helpful comments. In the following the original comments are inserted in *italic face* while our replies are printed in normal face.

**Reviewer 1:**

Page 3, Line 87: Figures should be referenced in numerical order: the first reference to a figure in the text should be to figure 1. (I would not re-order the figures to correct this, rather, I would remove the reference to figure 3 in this line.) Reply 1: Agreed and changed.

Figure 1 and Figure 3: The figures are really too small. They might be just about OK if made the full width of the two columns in the final version, but I am not sure about this. Text on figures should be a similar size to the text in the figure caption. The legends in these figures are particularly hard to read; it would help if they were not on top of the curves. **Reply 2:** Agreed. Figure 1 and figure 3 were split up in two individual figures which now span the full width of the page. The text was changed accordingly.

Figure 2: It is good that the two panels have the same colour scale for easy comparison. But the scale should go down to the lowest values shown in the left panel. As the figure stands, the colour white is used to represent two distinct things: negative values, and areas where there are no data. The "no data" areas should be left white, and the (white free) colour scale should extend so that it applies to the negative areas. The figure title "PAN, 200807" is not needed to distinguish the two panels and tells the reader nothing that is not in the caption. It should be removed. The contour lines should be in a colour (or colours) that allow them to be seen against the colour scale. With these changes made, Page 3 Line 113 will need to be changed (and can possibly be simplified). Many of these comments also apply to figures 4, 5 and 6.

**Reply 3:** Agreed. The colour scale was changed to cover the whole range of values of both figures. The titles have been removed. We did not change the contour line colour, because the background colours are now much lighter and white contour lines would be harder to see than black ones. The text was changed accordingly to reflect the updated colour scale.

Page 6 line 118: "setup" should be "set up" because it is a verb. (Note that on page 7 line 140, "setup" is a noun and should be left as it is.) **Reply 4:** Agreed and corrected.

Page 6 line 131: "The spectral region [...] could be narrowed to [values]" Narrowed from what? **Reply 5:** The gap could be narrowed down to 791.0 to  $792.0 \text{ cm}^{-1}$  from 790.5 to  $792.5 \text{ cm}^{-1}$ . Additional information on this is now given in the manuscript.

Figure 7: This figure is probably OK if shown at the full 8.3cm width of a journal column. The title "Rows of A" on each panel should be removed.

**Reply 6:** We changed the figures to better reflect the altitude region of interest. The titles were removed and the figures now span the entire width of the page.

Page 11 sec 5.1.1: It would be nice to add a figure showing the mean ATMOS profiles and a suitably-averaged MIPAS profile for the same time of year.

**Reply 7:** We now provide the suggested figure in the revised version of the manuscript. Average MIPAS profiles of the respective region and time of year are used. The text was changed accordingly.

Page 11 line 212: Remove comma after "profile". Reply 8: Agreed and removed.

Page 13 figure 9: It is again marginal whether this figure is large enough. It is probably OK at the full two-column width of the journal page. It would be a great improvement if the five panels were labelled (a) to (e) so that the text could refer to "panel (e)" rather than "second panel to the right". (I think this means "second panel from the right" but I am not sure; letter labels would remove this kind of confusion.)

Reply 9: Agreed and changed. The text was changed accordingly.

Figures 10 and 11: I again have concerns about the sizing of the figures. The text in the captions is unreadably small at the size of the review article. The title on Figure 11 is not needed. **Reply 10:** Agreed and changed. Fig. 10 was split up into two individual figures. These figures now each span the full width of the document. The text was changed accordingly.

**Reviewer 2:**

**Major comments**

The last sentence of the abstract should be removed, it states "The decline in CCl4 abundance during the MIPAS Envisat measurement period (July 2002 to April 2012) is clearly reflected in the retrieved distributions". I agree that information on (and a proper quotation of) the CCl4 trend would have been a very valuable addition to this study, BUT only a subset of the observations is presented, the periods shown do not cover the 10-year time interval (09/2003 - 04/2011 instead of 07/2002 - 04/2012) and the reader has no element to gauge the CCl4 rate of change and to judge about the validity of this assertion

**Reply 1:** Trends have now been estimated from the full data set and an respective figure was included in the paper. A subsection was added to discuss the results of the trend estimation. The according text states good agreement with the trends estimated by Valeri et al. (2017).

Figure 2 shows that the PAN product jointly retrieved with CCl4 is superior to the standard PAN data available thus far from the MIPAS team, it would be equally important to have an idea of the impact of retrieving versus neglecting PAN on the quality of the CCl4 product! In particular, is there a systematic impact on the CCl4 mixing ratios, allowing to close the well-known gap between in situ and remote-sensing data (see e.g. Chipperfield et al., ACP, 16, 2016)? This information would be very valuable for the community and I suggest adding two panels to Fig.2 dedicated to CCl4 with/without

**Reply 2:** We believe there is a misunderstanding here. None of the two figures show the PAN results for  $CCl_4$  being left out entirely in the retrieval.  $CCl_4$  was accounted for in the MIPAS retrieval before the gas was an actual target of the retrieval itself. However, optimizing the retrieval for  $CCl_4$  led to changes in the PAN distributions. The influence of these changes are reflected in the two panels of Fig. 2. Since both species, PAN and  $CCl_4$ , were accounted for in the original PAN retrieval, we do not see a benefit from showing  $CCl_4$  results without PAN. Fig. 2 is supposed to ensure that changes made to the retrieval to optimize it for  $CCl_4$  did not decrease the

quality of the PAN results. Fig. 2 proves that, on the contrary, these changes also led to improvement of the PAN results.

Section 4.2: it is somewhat strange that the FR measurements provide a lower DOF (3.5) than the RR observations (4.0). What could be the reason for this? This deserves a comment. **Reply 3:** We don't think this is strange, because the RR measurements have a finer altitude sampling. Measurements were taken at 27 instead of 17 tangent altitudes during the RR and FR period, respectively. This easily explains the higher DOF of the RR observations. A sentence was added for clarification.

Figure 7 is really small and the y-axis unnecessarily goes up to 80 km, I suggest limiting the altitude range to something like 0-50 km to improve readability **Reply 4:** Agreed. The altitude range was limited to 0-40 km and the figure is now spanning the full width of the page to improve legibility.

Section 5.1.1: ATMOS results are used for a qualitative comparison, but still, why did you use profiles retrieved in the mid-1980s by Zander et al, when the CCl4 spectroscopy was of poor quality? (see Brown et al., Appl. Opt., 35, 1996). Results reported later on by Zander et al. (e.g. GRL, 23, 1996) are very likely more appropriate for a sensible comparison. An alternative would be to use the ATMOS version 3 results available from

http://remus.jpl.nasa.gov/atmos/atmosversion3/atmosversion3.html and fully described in Irion et al. (Appl. Opt., 41, 2002)

**Reply 5:** Agreed. A figure regarding the qualitative comparison was added to the manuscript. We are now using results reported by Zander et al. (1996).

Section 5.2.1: the agreement between ACE and MIPAS is best below 15 km (lines 265-266 on page 13), but this is also mostly where the number of coincidences is the smallest (second left frame of Fig. 9). Could this inconsistent sampling have an impact on the statistics?

**Reply 6:** Since the comparison is based on coincident measurements, the impact of inconsistent sampling should be negligible.

**Minor comments and typos**

The title is not very informative; it could be edited to inform about the fact that first intercomparisons are included in this work

**Reply 7:** Agreed. The title was changed to "MIPAS IMK/IAA Carbon Tetrachloride (CCl4) Retrieval and first Comparison with other Instruments".

Page 2, line 22: "in 1987, when it was restricted": this is incorrect, CCl4 was not among the first species controlled under the Montreal Protocol, it was added to the list in the 1990 London Amendment **Reply 8:** Agreed and changed.

Page 2, line 28: these top-down emissions were evaluated instead of "reported" **Reply 9:** Agreed and changed.

Page 2, line 29, I think a comma is needed after "unreported"

Reply 10: Agreed and changed.

Page 2, lien 35, here, I suggest replacing "considerably" by "now" **Reply 11:** Agreed and changed.

Page 2, line 37: I would remove the reference to MIPAS here ("besides those of MIPAS...", it is appropriate to introduce the new measurements later on, after the review of previous works **Reply 12:** Agreed and changed.

Page 3, line 65, "as reduced" instead of "is reduced" **Reply 13:** Agreed and corrected.

Page 3, line 85: the information about the actual spectral range fitted to retrieve CCl4 is not consistent across the manuscript (see table 1, end of section 3.2...), this should be fixed **Reply 14:** Agreed and corrected.

Caption of Fig.2: I guess that the "Black: measured spectrum, hardly discernible because overplotted by modelled spectra" warning has nothing to do here... **Reply 15:** Agreed and removed.

Page 7, line 152: I would edit to "of CCl4 for different time periods. All of the..." Reply 16: Agreed and added.

Section 5.1.1.: ATMOS also participated to three other shuttle missions, in 1992, 1993 and 1994. Reply 17: Agreed. This information is now included in the text.

Section 5.2.: please reword to something like "Since all collocated measurements were retrieved using the spectroscopic data of Nemtchinov and Varanasi (2003) introduced in HITRAN 2000, MIPAS Envisat retrievals based on the same spectroscopic dataset were also used for consistency and in order not to mask possible other discrepancies." **Reply 18:** Agreed and changed.

**The following technical changes have been made with respect to the accepted version:**

- Since Fig. 13 was slightly blurred the figure was replotted. It is showing exactly the same data as in the originally submitted version of the manuscript.
- The legends of Fig. 14 and 15 were rearranged to repositioned in for better legibility and no abbreviations are used any more.

[revised manuscript text omitted]

The spectral regions used for the retrieval of CCl4 are 772.0 - 791.0 cm-1 and 792.0 - 805.0 cm-1. The gap from 791.0 to 792.0 cm-1 is necessary, since even when accounting for line mixing, strong effects from the CO2 Q-branch still <del>occurred</del> <del>occur</del> in the residuals<del>(Fig. ??, right plot). Several results</del> from previous steps in the retrieval chain were used to derive CCl4 (Table 1) including the spectral shift ( $z_{tangent}$ ), the temperature (T), the horizontal temperature gradient (Tgrad) and mixing ratio profiles of HNO3, ClO, CFC-11, C2H6,

**Figure 1.** Examplary spectra of MIPAS CCl4 at 12 km and 11.5 km, respectively. Left: during the FR period (September 2003). Right: RR period (July 2008). Top panelspanel: spectra; bottom panelspanel: residuals.

HCN, ClONO2 and HNO4.

In addition, several species were found to improve the retrieval whenever their mixing ratio profiles were fitted alongside CCl4. These are peroxyacetyl nitrate (PAN), CH3CCl3,

- 155 side CCl4. These are peroxyacetyl nitrate (PAN), CH3CCl3, HCFC-22, O3, H2O, C2H2 and COF2. Although for most of 175 these species results from preceding retrieval steps are available, fitting their concentrations jointly with that of CCl4 reduces the fit residuals significantly. This is attributed to spec-
- troscopic inconsistencies of the interferers' spectroscopic data between the spectral region where these were retrieved and the spectral region where CCl4 is analyzed. Also fitted were a background continuum accounting for spectral contri-180 butions from aerosols and a radiance offset which is constant
   for all tangent altitudes (Table 1).
- These specifications retrieval settings lead to spectral fits as displayed in Fig. **??**1 and Fig. 2, where an example for the FR period (left) and the RR period (right) are shownare shown, 185 respectively. The measured spectra are plotted in black (not discoursible from the best fitting for modelled in black (not
- discernible from the best fitting fit modelled in the fitting

window), while the red and the blue lines represent the modelled spectra of the regions from  $772.0 - 791.0 \text{ cm}^{-1}$  and  $792.0 - 805.0 \text{ cm}^{-1}$ , respectively. Some periodic residuals are visible in both the FR and the RR period. These result from less than perfectly fitted CO2, but as will be shown in Sec. 5, are only of minor relevance for the accuracy of the retrieved CCl4.

**3.1 Information cross-talk with PAN**

The signature of PAN is particularly prominent in the spectral region of  $CCl_4$  and can thus be retrieved during the same retrieval step. Actually, jointly fitting PAN improves is very important for the  $CCl_4$  retrieval. Since PAN was already retrieved from MIPAS spectra before (Glatthor et al., 2007), it is of obvious interest to investigate the PAN results from the  $CCl_4$ -PAN joint retrieval in comparison with those from the original PAN retrieval. In there  $CCl_4$  was fitted alongside PAN but the retrieval was not yet optimized for  $CCl_4$ .

We find slightly higher volume mixing ratios of PAN